# Last glacial inception trajectories for the Northern Hemisphere from coupled ice and climate modelling

Taimaz Bahadory[1], Lev Tarasov[1], and Heather Andres[1]

[1]Memorial University of Newfoundland and Labrador

**Correspondence:** Lev Tarasov (lev@mun.ca)

**Abstract.** We present an ensemble of Last Glacial Inception (LGI) simulations for the Northern Hemisphere that captures a significant fraction of inferred ice volume changes within proxy uncertainties. This ensemble was performed with LCice 1.0, a coupled ice sheet and climate model, varying parameters of both climate and ice sheet components, as well as the coupling between them. Certain characteristics of the spatio-temporal pattern of ice growth and subsequent retreat in both North America (NA) and Eurasia (EA) are sensitive to parameter changes while others are not. We find that the initial inception of ice over NA and EA is best characterized by the nucleation of ice at high latitude and high elevation sites. Subsequent spreading and merger along with large-scale conversion of snow fields dominate in different sectors. The latter plays an important role in the merging of eastern and western ice regions in NA.

The inception peak ice volume in the ensemble occurs approximately at 111 ka and therefore lags the summer 60°N insolation minimum by more than 3 kyr. Ice volumes consistently peak earlier over EA than NA. The inception peak in North America is characterized by a merged Laurentide and Cordilleran ice sheet, with Davis Strait covered in ice in ∼80% of simulations. Ice also bridges Greenland and Iceland in all runs by 114 ka and therefore blocks Denmark Strait. This latter feature would thereby divert the East Greenland Current and Denmark Strait overflow with a potentially significant impact on ocean circulation. The Eurasian ice sheet at its inception peak varies across ensemble runs between a continuous ice sheet to multiple smaller ice caps.

In both continents, the colder high latitudes (*i.e.* Ellesmere and Svalbard) tend to grow ice through the entire simulation (to 102 ka), while lower latitudes lose ice after ∼110 ka. We find temperature decreases over the initial phases of the inception lead to the expansion of NA ice sheet area, and that subsequent precipitation increases contribute to its thickening. EA ice sheet area also expands with decreasing temperatures, but sea ice limits any increases in precipitation, leading to an earlier retreat away from the EA maximum ice sheet volume.

We also examine the extent to which the capture of both LGI ice growth and retreat constrains the coupled ice/climate model sensitivity to changing atmospheric $pCO_2$. The 55 member sub-ensemble that meets our criteria for "acceptable" ice growth and retreat has an equilibrium climate sensitivity lower bound that is 0.3°C higher than that of the full ensemble. This suggest some potential value of fully coupled ice/climate modelling of the last glacial inception to constrain future climate change.

## 1 Introduction

Reconstructions of sea level change from corals and oxygen isotope records (*e.g.* Waelbroeck et al., 2002; Siddall et al., 2003) along with some limited inferences from glacial geology (Clark et al., 1993) indicate that between about 120 and 115 ka, large ice sheets formed rapidly in the Northern Hemisphere (NH). By 110 ka, mean sea level is inferred to have been approximately 45-65 m lower than present (Lambeck and Chappell, 2001; Waelbroeck et al., 2002; Siddall et al., 2003; Lisiecki and Raymo, 2005) or about half of that inferred for LGM. Contrary to the common perception that ice sheet growth is a much slower process than ice sheet retreat, this large last glacial inception (LGI) growth in ice volume occurred over approximately the same duration (∼10 kyr) as the last deglaciation. This rapid ice sheet growth was subsequently followed by ice retreat for the next 10 kyr (Bard et al., 1990; Chappell et al., 1996; Gallup et al., 2002).

Aside from global constraints on sea level, little is known about the LGI evolution of individual ice sheets. The terrestrial geological record was largely destroyed by subsequent ice advance and retreat, and any proxy records that may have survived are scattered and have large age uncertainties (Andrews and Barry, 1978; Lambeck and Chappell, 2001; Stokes et al., 2012b; Batchelor et al., 2019). This uncertainty percolates into the associated changes in the climate system (especially over terrestrial sectors): due both to similar limitations in proxy records for climate characteristics and uncertainties in the required ice sheet boundary conditions for running climate models over this interval.

Given the rapidity of LGI sea level decreases and the relative sizes of last glacial maximum ice sheets, it is generally assumed that North America contributed a significant fraction to this sea level fall. The rapid ice growth has also motivated the development of one hypothesis to characterize glacial inception over North America: widespread thickening of snowfields (Andrews and Mahaffy, 1976). A second complementary hypothesis stems from consideration of present-day mid-latitude glaciers and posits ice sheet spreading from high elevation nucleation sites (Weertman, 1964). A previous attempt to simulate the inferred sea level drop during LGI supported the widespread snowfield thickening paradigm (Calov et al., 2005a). The model used in that study employed a very low grid resolution (51°longitude by 10°latitude for the atmosphere and approximately 100 km for the ice sheet model) and presented only three transient simulations. Given the uncertainties in the proxy data and models, a much larger ensemble of simulations that better captures model uncertainties is required to assess how representative this result is of the actual growth of ice sheets during the LGI.

Ideally, model studies of LGI would employ sophisticated Earth System Models (ESMs) at high resolution bidirectionally coupled to ice sheet models to produce ensembles of transient experiments that span the uncertainties of the relevant data and processes, but this is computationally too expensive. Instead, model studies of LGI tend to make one of two simplifications. First, general circulation model- (GCM) based studies treat the climate in a sophisticated way, but rely on a small number of snapshot experiments without interactive ice sheets. Ice sheet boundary conditions are prescribed, which can lead to a modelled climate that is inconsistent with the prescribed ice extent (Pollard and PMIP-participating groups, 2000). Furthermore, the reliance of these studies on at most a few model runs severely limits any possible uncertainty assessment. Second, experiments performed with ice sheet and climate models coupled together tend to employ Earth System Models of Intermediate Complexity (EMICs). These model configurations include interactive ice sheets and can be run with transient boundary conditions.

However, their low climate model resolution means more processes must be highly parameterized, and some key ice/climate feedbacks are not modelled at all.

Due to such simplifications, most LGI model studies have been unable to simultaneously simulate the required rapid ice build-up until around 110 ka with the subsequent retreat (*e.g.* Tarasov and Peltier, 1997a; Wang et al., 2005; Calov et al., 2009; Bonelli et al., 2009). Prior to the development of the LCice 1.0 (Bahadory and Tarasov, 2018), the one model that has adequately captured both the growth and retreat phases of LGI is CLIMBER-2. With this model, Ganopolski et al. (2010) used an imposed (albeit plausible) aeolian dust deposition forcing on snow albedo and a temperature bias correction to capture LGI as well as the whole glacial cycle. An earlier attempt (Kageyama et al., 2004) using a similar version of CLIMBER (though with a different ice sheet model) but without bias correction, no dust forcing, and a simple positive degree surface mass balance scheme was only able to glaciate 19 mESL by 108 ka, with insignification subsequent retreat by the run termination at 106 ka. In a more recent configuration of CLIMBER-2, Willeit and Ganopolski (2018) used a dynamical aeolian dust model to again approximately replicate the last glacial cycle sea level record (including capturing LGI). Their work demonstrated the critical role of dust deposition in their model: without dust deposition, ice volume grew monotonically until about 90 ka to an ice volume of over 300 m in sea level equivalent (SLE). When run over 4 consecutive glacial cycles, a somewhat differently configured version of CLIMBER-2 had a much weaker post-LGI retreat (Ganopolski and Brovkin, 2017). However, it should be noted that the latter study was able to capture the larger-scale features of the last 4 glacial cycles with the model only subject to orbital forcing (*i.e.* with internally computed greenhouse gases and dust load), a feat yet to be replicated by any other ESM. The CLIMBER EMIC was designed to simulate multiple glacial cycles while striving for inclusion of all relevant earth system components. This necessitated trade-offs in model resolution and levels of approximation (such as the use of a 3 basin 2D ocean model). The question remains whether dust loading would play as important a role in replicating the last glacial cycle in other models (with different trade-offs) as it did in CLIMBER-2.

Attempts to capture LGI with ice sheet models coupled to more advanced general circulation climate models have to date only attempted to capture Eemian interglacial to stadial growth (and not the subsequent interstadial ice retreat). Published attempts to date have failed to attain adequate ice growth rates even with constant orbital forcing chosen to minimize summer insolation over ice sheet regions. Herrington and Poulsen (2011) using the GENESIS Atmospheric GCM at T31 with a slab ocean only attained a 15 mESL growth in North America ice after 10 kyr of constant 116ka orbital forcing, likely in part due to the large 500 year asynchronous coupling timestep. Gregory et al. (2012) using the FAMOUS AOGCM only attained 3.9 and 1.6 mESL for North America and Eurasia respectively after 10 kyr, with constant 115 ka orbital forcing and 290 ppmv pCO2.

Temperature bias corrections are also somewhat problematic, even though they are ubiquitous in coupled ice sheet and climate modelling. They rely on the standard (though often implicit) justification that climate models are more likely to better capture the perturbative response to radiative forcing changes than the actual present-day temperature distribution. Whether this assumption adequately holds for perturbations as large as during glacial stadials is unclear. Furthermore, bias corrections are generally imposed externally to the climate model, so the glacial climate imposed on the ice sheet model is dynamically inconsistent.

In order to test the necessity of both temperature bias corrections and any form of parametrized dust impact on surface mass balance, we have chosen for this initial investigation to avoid both interventions. Instead, LCice 1.0 includes all the main feedbacks between the ice sheet and the atmosphere and ocean, many of which have not been resolved in previous coupled EMIC/ice sheet modelling studies (Bahadory and Tarasov, 2018). As a result, LCice 1.0 is so far the only fully coupled ice sheet-climate model demonstrably capable of approximately simulating both the rapid growth and retreat phases of the LGI (Bahadory and Tarasov, 2018) without using any bias correction or imposed dust forcing. It is also fast enough to generate ensembles of glacial cycle timescale transient simulations.

We employ LCice 1.0 in this study to generate an ensemble of transient LGI simulations and address the following questions. How did each ice sheet most likely evolve through its inception phase, and which of the two aforementioned paradigms best describes this evolution? More fundamentally, is the spatio-temporal pattern of LGI ice a single attractor in the trajectory space of possible glacial inceptions, or could small changes in initial conditions or physical properties (*e.g.* snow albedo) lead to a significantly different pattern? More crudely, did the LGI have to happen the way it did? Addressing this last question includes an examination of the extent to which the evolution of ice sheets in Eurasia (EA) and North America (NA) are correlated. Expanding this trajectory space analysis to the climate, we also examine how the climate conditions (insolation, carbon dioxide, temperature and precipitation) facilitate or hinder the rapidity of ice growth and retreat.

The capture of LGI ice growth and subsequent decay potentially constrains the sensitivity of coupled ice and climate models to projected future increases in $pCO_2$, as the largest sources of uncertainty in such coupled models are the internal climate system feedbacks and not the much more tightly constrained direct radiative forcing of changing atmospheric $pCO_2$. To test this hypothesis, we also examine the extent to which capturing the LGI constrains the Equilibrium Climate Response (ECR) of LCice to a doubling of atmospheric $pCO_2$.

In section 2, we first review LCice 1.0 and its components, and the choice of our parameters for the ensemble study. We discuss the phasing of LGI in our ensemble in section 3 in terms of ice sheet and climate evolution. The implications of our results for ice/climate model sensitivity are discussed in section 4.7.

## 2 Experimental setup

We ran an ensemble of 500 simulations for the North American, Greenland and Eurasian ice sheets using the coupled model LCice 1.0. These 500 simulations were previously sieved from a larger ensemble of 2000 simulations covering the preindustrial to present day interval. Only 55 out of 500 inception simulations could approximately replicate the pattern of sea level lowering due to ice sheet build up, followed by sea level increase, as suggested by reconstructed proxies of Waelbroeck et al. (2002) and Lisiecki and Raymo (2005).

In detail, the acceptance criteria for the 55 "acceptable" simulations were: 1) at least a 24 m eustatic sea level contribution to the LGI sea level minimum from ice sheet growth and 2) at least an 8% subsequent increase in eustatic sea level by 105 ka. The rejected simulations generally underestimated total ice volume, though a small number of simulations captured appropriate

growth without a subsequent retreat phase. For the rest of this paper, the term "ensemble" refers to this sieved group of 55 simulations.

## 2.1 Ensemble parameters and sensitivity analysis

The ensemble is constructed by varying 18 parameters, 5 of which are found in LOVECLIM, 9 in the GSM, and 4 in the coupler, as described in (Bahadory and Tarasov, 2018). The LOVECLIM ensemble parameters include snow albedo, bare-ice

albedo, melting ice albedo, the humidity threshold for parameterized precipitation, and the cloud parameterization scheme. The GSM ensemble parameters address uncertainties in basal drag, ice calving, sub-shelf melt, and deep geothermal heat flux. Ensemble parameters related to the coupling procedure include spinup length and start time, upscaling method, and the method used to calculate the vertical temperature gradient. Each ensemble parameter and associated sensitivity analysis for the coupled model is described in detail in Bahadory and Tarasov (2018).

## 2.2 Initial conditions and spin-up

Since the extent of the Greenland ice sheet during the Eemian is not well constrained, the initial state of the ice sheet at the start of all simulations is set to its present-day configuration. Future work will use an initialization from ongoing Greenland ice sheet model calibration. The initial climate state is provided by a 3 to 5 kyr LOVECLIM spinup under transient Eemian orbital and greenhouse gas forcing, with present-day topography and ice mask.

## 2.3 Models

### 2.3.1 LOVECLIM

LOVECLIM is a coupled EMIC, consisting of a quasi-geostrophic atmosphere (ECBilt), a primitive equation ocean with dynamic sea ice (CLIO) and dynamic vegetation (VECODE). The horizonal resolution of the 3 level atmospheric component is T21. The ocean and sea ice components each have a resolution of 3°. LOVECLIM is fast enough to simulate LGI (120 ka to

145 100 ka) in less than 3 weeks using a single commodity core. It has therefore been used to simulate a wide range of different climates from the LGM (Roche et al., 2007) through the Holocene (Renssen et al., 2009) and the last millennium (Goosse et al., 2005) to the future (Goosse et al., 2007). It has also been used to recently model marine isotope stage 7 (Choudhury et al., 2020) in a fully coupled ice/climate model configuration, though with present-day bias corrections on temperature and precipitation.

Interpreting model-based results always requires a cognizance of model limitations. Aside from the simplified atmospheric dynamics and low grid resolution, a key limitation of LOVECLIM for our study is the fixed land-ocean mask. With an inferred LGI maximum sea level drop of approximately 45-65 m, throughflow through ocean gateways can change significantly (including the complete closure of Bering Strait). LOVECLIM is unable to handle a changing land mask except for in the Bering Strait, where throughflow is parameterized as a function of modelled sea level and regional ice sheet cover. Other potentially

important factors which can affect the results include simplified radiation and hydrology schemes, and the missing feedbacks of atmospheric dust on radiative forcing and surface mass balance.

### 2.3.2 GSM

The glacial systems model (GSM) is built around a thermo-mechanically coupled ice sheet model. It includes a 4 km deep permafrost-resolving bed thermal model (Tarasov and Peltier, 2007), fast surface drainage and lake solver (Tarasov and Peltier, 2006), visco-elastic bedrock deformation (Tarasov and Peltier, 1997b), Positive Degree Day surface mass balance with temperature dependent degree-day coefficients derived from energy balance modelling results (Tarasov and Peltier, 2002), sub-grid ice flow and surface mass balance for grid cells with incomplete ice cover (Morzadec and Tarasov, 2017), and various ice calving schemes for both marine and pro-glacial lake contexts (Tarasov et al., 2012). For the results herein, ice shelves are treated using a crude shallow ice approximation with fast sliding. The GSM runs at 0.5°longitude by 0.25°latitude grid resolution.

The largest internal source of error in the GSM is the crude treatment of ice shelves (which has been rectified in the latest version of the GSM). Marine sectors are also problematic due to both the unavoidable use of a (potentially distant) upstream water temperature profile to drive subshelf melt and the lack of an efficient well-constrained model for ice calving in the community. The combination of these three sources of uncertainty will impact ice shelf extent and grounding line position. Extrapolating from the results of a comparison of grounding line response sensitivity to changes in ice rheology for different ice dynamical approximations (Pattyn et al., 2012), it is likely that the GSM underestimates grounding line response.

Another potential source of error is our use of a PDD scheme for surface melt, which, as is typical, does not include explicit dependence on surface insolation. A few recent studies have drawn attention to the direct impact on surface mass balance by changing insolation from orbital forcing for glacial cycle scale contexts (van de Berg et al., 2011; Bauer and Ganopolski, 2017). These studies have unfortunately only invoked comparisons against simplistic PDD schemes with constant melt coefficients. As such, it is unclear whether our scheme has the same magnitude of error (and associated ice sheet impact) under different orbital forcing.

### 2.3.3 LCice 1.0 coupler

The LCice coupler is designed to extract, regrid, and exchange the required fields between atmosphere and ocean components of LOVECLIM and the GSM asynchronously (*i.e.* LOVECLIM and the GSM are run sequentially with boundary conditions from the other model fixed between data exchanges). On the basis of sensitivity tests, the time between data exchanges was chosen to be 20 years as the optimal balance between efficiency and proximity to shorter coupling time step solutions (Bahadory and Tarasov, 2018).

Fields passed from the ice sheet to the atmosphere include ice mask and surface elevation, the latter via one of the three included schemes (simple, envelope, and silhouette, the choice of which is under ensemble parameter control). The atmosphere to ice coupling includes the monthly mean and standard deviation temperature and monthly mean precipitation, evaporation, wind direction and magnitude, and vertical temperature lapse-rate. LCice 1.0 uses an innovative scheme to downscale precipitation to the ice model grid that accounts for orographic forcing on the GSM grid resolution topography. Temperature downscaling

uses the evolving vertical surface temperature gradient field of LOVECLIM. The coupler also includes a simple radiative cloud parameterization to compensate for the present-day prescribed radiative cloud cover of LOVECLIM.

In ice sheet-ocean interactions, the GSM determines the runoff routing and passes freshwater fluxes to the ocean model. The ocean model provides the GSM with vertical temperature profiles, which are required to calculate sub-shelf melt. Details of each component of the coupling and their influence are described in Bahadory and Tarasov (2018).

Given model limitations, there is no one best run in the ensemble. Instead, different runs have different features, each of which will likely have different patterns of misfits against inferred proxy records. In the following results, we crudely interpret

feature frequency in the ensemble to be a partial metric of feature likelihood, though this is far from a rigorous probabilistic analysis.

## 3    Results

The total Northern Hemisphere ice volume averaged over the ensemble of 55 runs that meet our acceptance criteria (described above) is plotted in figure 1. No single ensemble parameter determines which runs meet the filter condition (not shown).

The maximum ice volume achieved by the LCice 1.0 ensemble during inception is much less than that inferred by Lisiecki and Raymo (2005), but within the collective uncertainty of the three reconstructions presented here (Waelbroeck et al., 2002; Siddall et al., 2003; Lisiecki and Raymo, 2005). The ensemble mean maximum ice volume is about 12 m SLE short of the Red Sea record (Siddall et al., 2003, short-dashed purple line in figure 1). This under-estimation is likely due in part to the absence of any contribution from the Antarctic ice sheet (and perhaps Patagonian and Tibetan ice caps). It is also consistent

with the fact that the simulated ice sheet volumes never reach the peak rate of ice growth indicated by any of the sea level reconstructions.

The timing of when the LCice 1.0 simulations achieve their maximum inception ice sheet volume is bounded by the three proxy-based reconstructions shown in figure 1. All but the Greenland ice sheets reach their maximum LGI ice volumes at least 3 kyr after the 60°N summer insolation minimum (orange line in figure 1). The earliest retreat occurs in the Red Sea

reconstruction. This reconstruction suggests a faster decrease in pre-stadial sea level compared to that of the other three records, and its timing of the sea level minimum and subsequent sea level rise is slightly advanced of the LCice ensemble mean. The LCice maximum ice sheet volume occurs approximately midway between the timing of minimum insolation at 60°N and minimum $pCO_2$. The Lisiecki and Raymo (2005) stadial peak occurs 2 kyr later, approximately halfway between the 60°N JJA (mean June July August) orbital minimum at 114.5 ka and the subsequent maximum at 104 ka.

A second test of the representativeness of these simulations for the LGI is made between temperature changes from a glaciological inversion of the GRIP ice core $\delta^{18}O$ record (Dansgaard et al., 1993; Tarasov and Peltier, 2003) and annual-mean temperatures calculated from the model grid cell containing its location. The ensemble mean 2m temperature anomaly relative to 119 ka follows the general trend of GRIP reconstructed temperatures in figure 2 until $\sim$ 112 ka. Individual runs have higher decadal to centennial scale variance than that of the GRIP record. However, the large millennial scale variability

of the GRIP record inversion is not captured by the simulations. The ensemble-mean annual temperatures from the GRIP

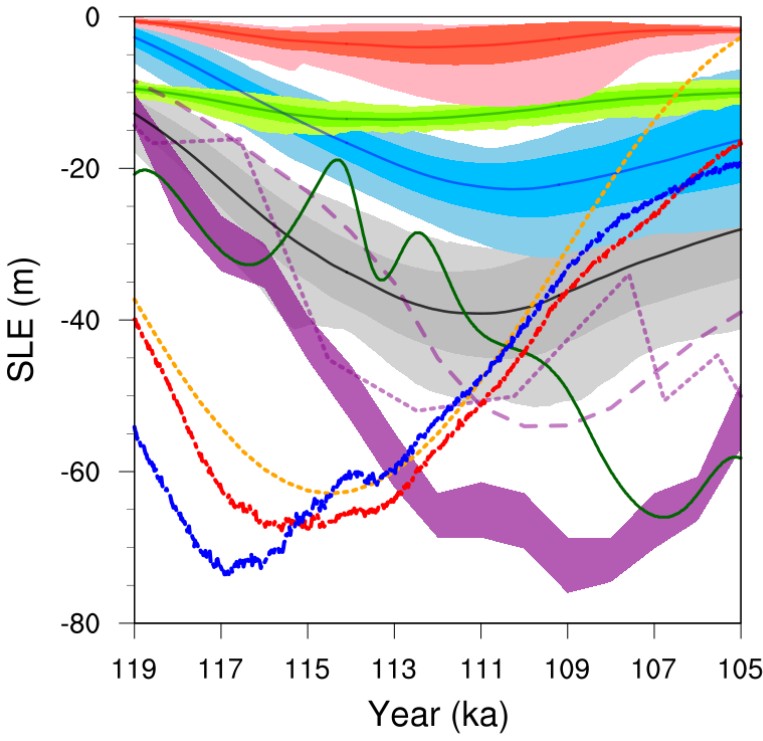

**Figure 1.** The time evolution of total (black), NA (blue), EA (red), and Greenland (green) ensemble mean ice volumes in m sea level equivalent (SLE) between 119 and 105 ka. The dark shading indicates the ±1 standard deviation range around the mean. The light shading shows the range between minimum and maximum ice volumes in the ensemble. The purple shading and lines (long-dashed and short-dashed) show the respective proxy-based sea level reconstructions from Lisiecki and Raymo (2005) with 1 sigma, Waelbroeck et al. (2002), and Siddall et al. (2003). To make the sea level reconstructions commensurate with the ice volumes, the present-day ice volume of Greenland has been added to them. The orange and dark green lines depicts the timing of insolation changes at 60°N and $pCO_2$, respectively. The JJA ensemble mean temperatures over 50°N-65°N of NA and 60°N-75°N of EA are shown as thick-dotted blue and red lines, respectively.

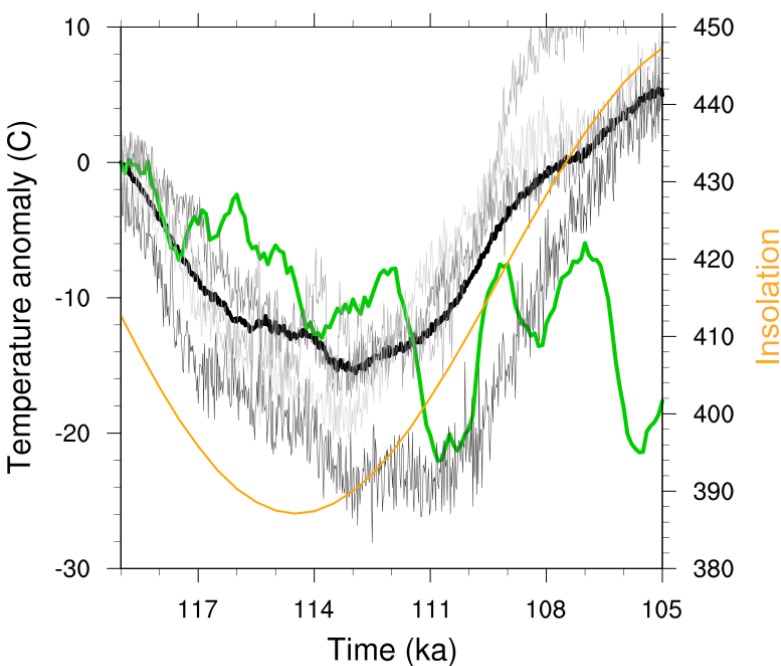

**Figure 2.** Annual mean 2 m temperature anomaly relative to 119 ka for the GRIP ice-core (green) (Dansgaard et al., 1993; Tarasov and Peltier, 2003), ensemble mean (thick black), and three individual runs (gray lines). The orange line depicts the timing of insolation changes at 60°N.

site subsequently diverge from reconstructed temperatures after approximately 111 ka. At this time, simulated temperatures increase at the GRIP site following insolation changes, whereas there is no evidence of a similar increase in the GRIP record temperature inversion. Instead, reconstructed GRIP temperatures exhibit multi-millennial timescale oscillations around stable, stadial (cold state) temperatures. It is unclear what mechanism would sustain stadial temperatures over central Greenland under
increasing insolation, especially since the simulations consistently predict that strong warming should result. It may be that this discrepancy reflects in part a lack of accounting for at least two standard sources of uncertainty in water isotope to temperature inversions: changes in the moisture source region and changes in the seasonal distribution of precipitation.

## 3.1 Glacial inception trajectory space

Having established that LCice 1.0 is able to capture both the ice sheet growth and retreat phases of the LGI, we explore the
pattern(s) of the ice growth and retreat across ensemble members. We start by analyzing the spatial patterns of EA and NA ice sheets at two diagnostic time intervals: first, the early stage of ice build up, and second, during the peak of the inception around

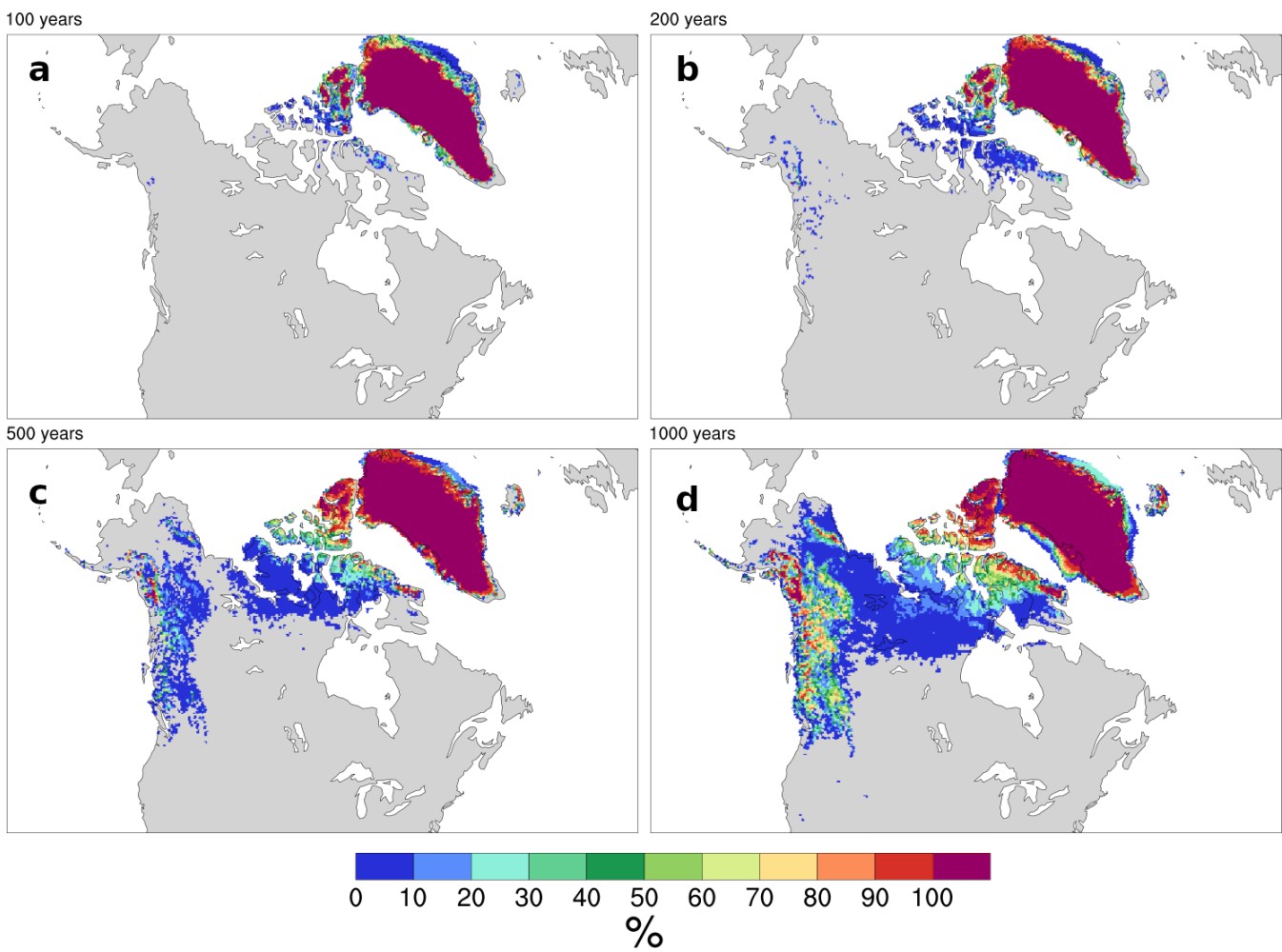

**Figure 3.** Percent of runs exhibiting ice cover in each grid cell for NA after **a** 100, **b** 200, **c** 500, and **d** 1000 years of simulation.

112 ka. Next, we explore the consistency of ice and climate evolution between these two intervals and during the subsequent retreat phase.

### 3.1.1  Spatial pattern of first appearance of ice

Despite having different start times (due to different calendar start years between 122 ka and 119 ka and spinup lengths varying between 3 to 5 kyr), all simulations start growing ice in the first 100 years of simulation (see figures 3.a and 4.a). Therefore, we analyze the spatial patterns of the first appearance of ice in the first 1000 years of simulation, rather than aggregating simulations according to a common calendar year.

In NA, all runs have extensive glaciation over Ellesmere and eastern Devon Island after 100 years of transient simulation 240 (figure 3). Subsequently, ice starts to spread through the Arctic archipelago and Baffin Bay sector of Baffin Island. This is in

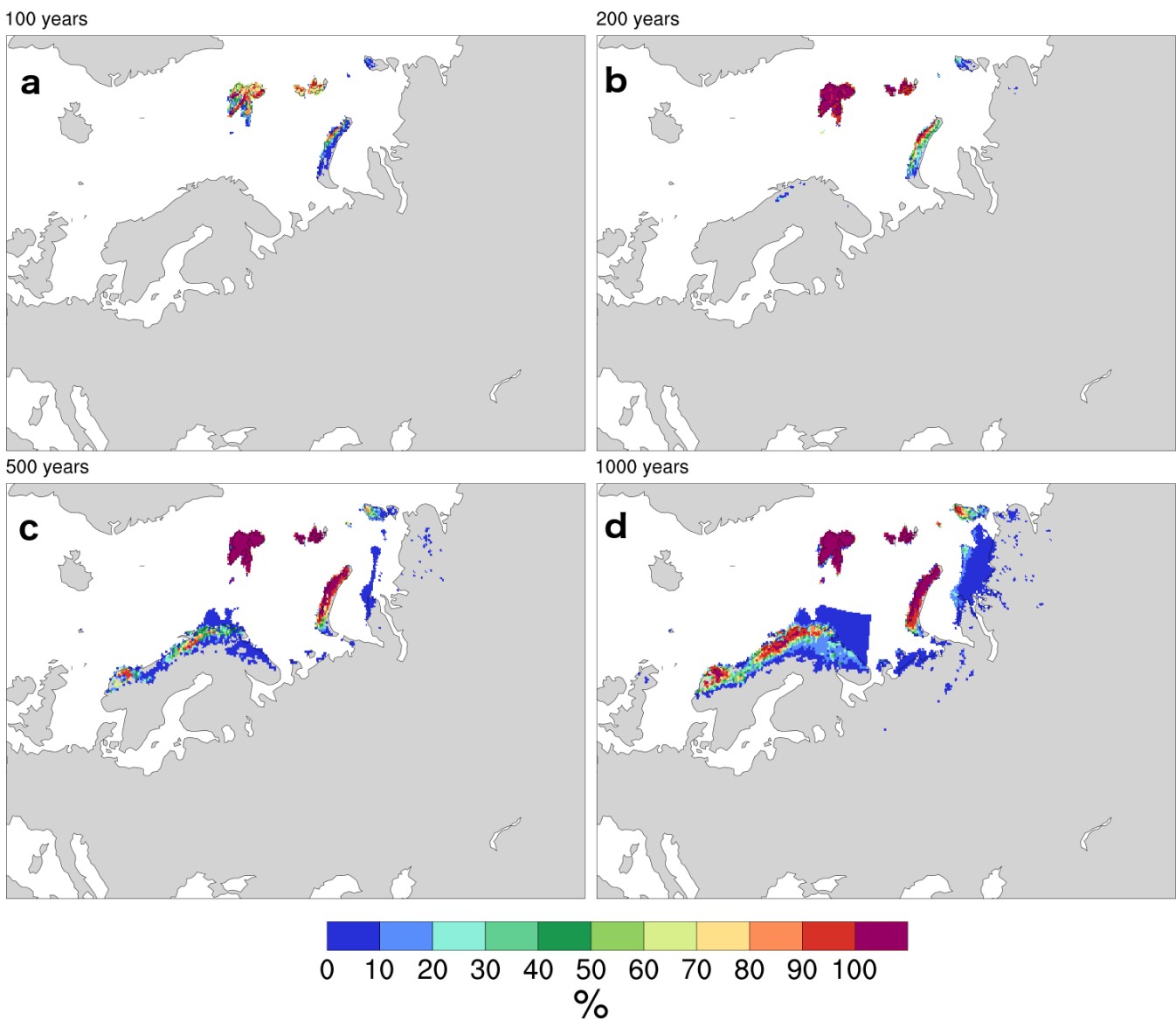

**Figure 4.** Percent of runs exhibiting ice cover in each grid cell for EA after **a** 100, **b** 200, **c** 500, and **d** 1000 years of simulation.

agreement with past suggestions that the first ice nucleation in NA occurs over the Canadian Archipelago with further growth, merger, and then expansion to southern and western regions (Weertman, 1964). This result is also consistent with the ongoing presence of extensive glaciers and small ice caps in this region.

By 1000 years, more than 20% of runs have extensive ice over the Pacific Cordillera down to 48°N. Northwestern Alaska remains ice free for the first 1000 years in all runs as does the non-Cordilleran sector of NA below 61°N.

To get a more detailed sense of what glacial inception might look like, it is worth examining ice sheet evolution for one of the runs that best fit sea level proxies. By 119 ka, most of NA above 65°N has ice cover, though much of it with surface elevation less than 500 masl (figure 5). The Canadian Cordilleran at this time is nearly completely covered with ice, particularly in locations above 1000 masl.

Ice nucleation over EA starts over the high precipitation and higher elevation Norwegian and Barents Sea sectors. Within the first 100 years of simulation, all runs exhibit ice growth over Svalbard, while some runs also show ice cover over other islands in the region (figures 4.a and b). After 200 years, Svalbard and Franz Joseph Land have complete ice cover in almost all runs, while Fennoscandia has no ice in almost all runs. By 500 years, nearly all runs have covered most of Novaya Zemlya in ice as well. Fewer than 10% of runs have any ice over Continental Russia during the first 1000 years.

Note that in figure 4, parts of the Fennoscandia ice margins in the Barents Sea follow unphysical, straight lines. This is an artifact of the model setup for submarine melt and is discussed in more detail in the Discussion.

### 3.1.2 Spatial pattern of the Last Glacial Inception maximum ice

To capture the maximum in ice volume for EA and NA during the LGI, we consider time slices for 114 ka, 112 ka, 110 ka and 108 ka in figures 6 and 7. We aggregate our simulation results according to their boundary condition years rather than their simulation years.

At 114 ka, the Cordilleran is completely ice covered in all runs down to approximately 45°N. Central NA ice extends to approximately 55°N until a sharp northward turn of the southern ice margin over James Bay extending to the east (figure 6). Labrador and eastern NA remain ice-free, likely due to warm model biases in this region (*cf.* discussion below). The Greenland and Iceland ice sheets are bridged by ice across Denmark Strait in all runs by 114 ka (with most runs having grounded ice right across the Strait). Alaska is almost fully ice-covered in all of the simulations.

The main differences in peak LGI NA ice extent between ensemble members occur at the northwestern Alaskan ice margin (40% of ensemble runs cover Bering Strait at 114 ka), at the southern margin, and over Davis Strait. For the latter, approximately 80% of simulations create an ice bridge connecting the Laurentide and Greenland ice sheets across the Strait. This ice bridge generally starts out from a merger of opposing ice shelves. For some (but not all) ensemble runs, it can also ground right across the Strait and therefore isolate Baffin Bay from the Labrador Sea.

After the stadial peak in NA ice volume, the main variation between ensemble members appears in the rate of ice retreat. Initially, while the south-eastern ice margin rapidly retreats to higher latitudes in simulations with smaller ice sheets, simulations with larger ice sheets show little change in ice extent. This difference in behaviour leads to the largest difference in ice extent over Hudson Bay at 110 ka, when the entire area is ice-covered in approximately 20% of the simulations while 30%

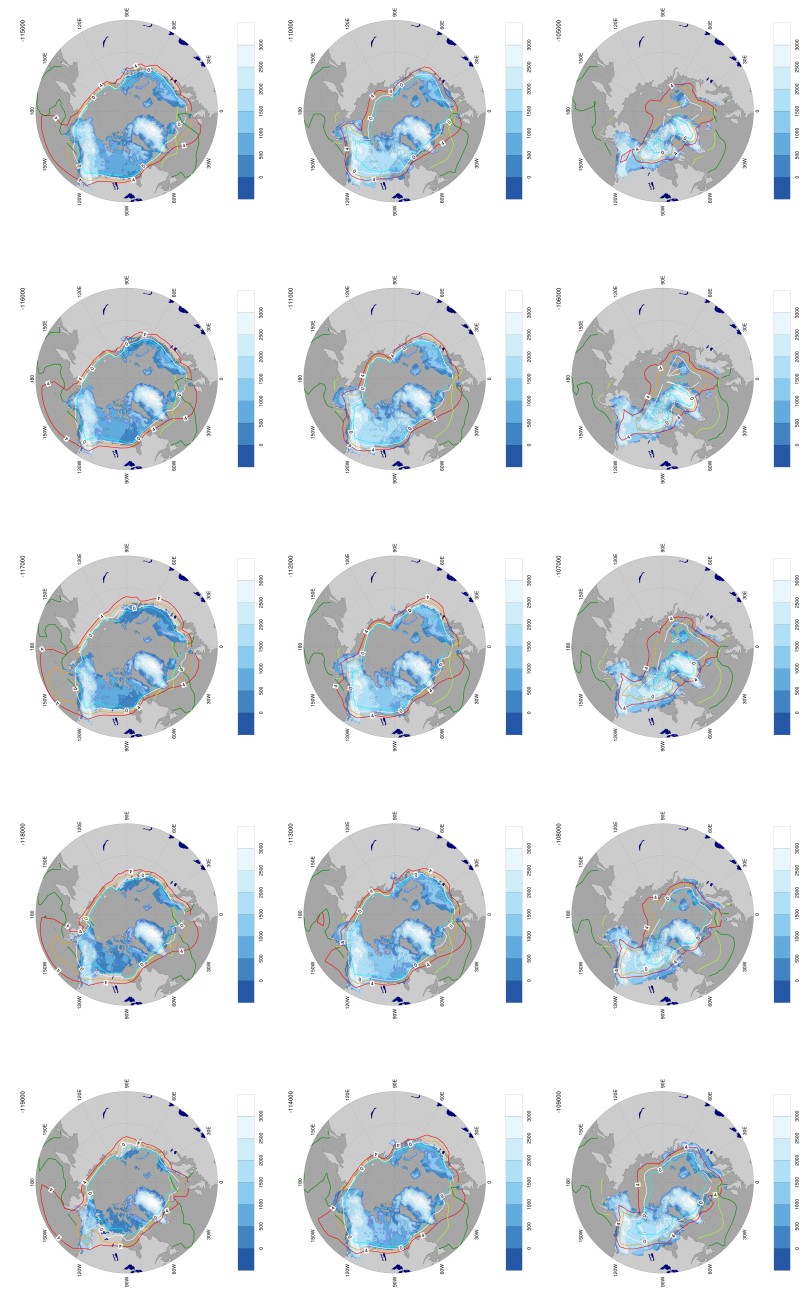

**Figure 5.** The evolution of ice sheet surface elevation (shaded areas in light blue–white gradient), 2 meter JJA temperature ($-2$°C to 4°C ), and sea ice seasonal maximum and minimum extent (dark and light green) for every 1 kyr from 119 ka to 105 ka for one of the best fitting (to proxy sea level records) simulations of the ensemble. The 1000 m elevation contour is in purple.

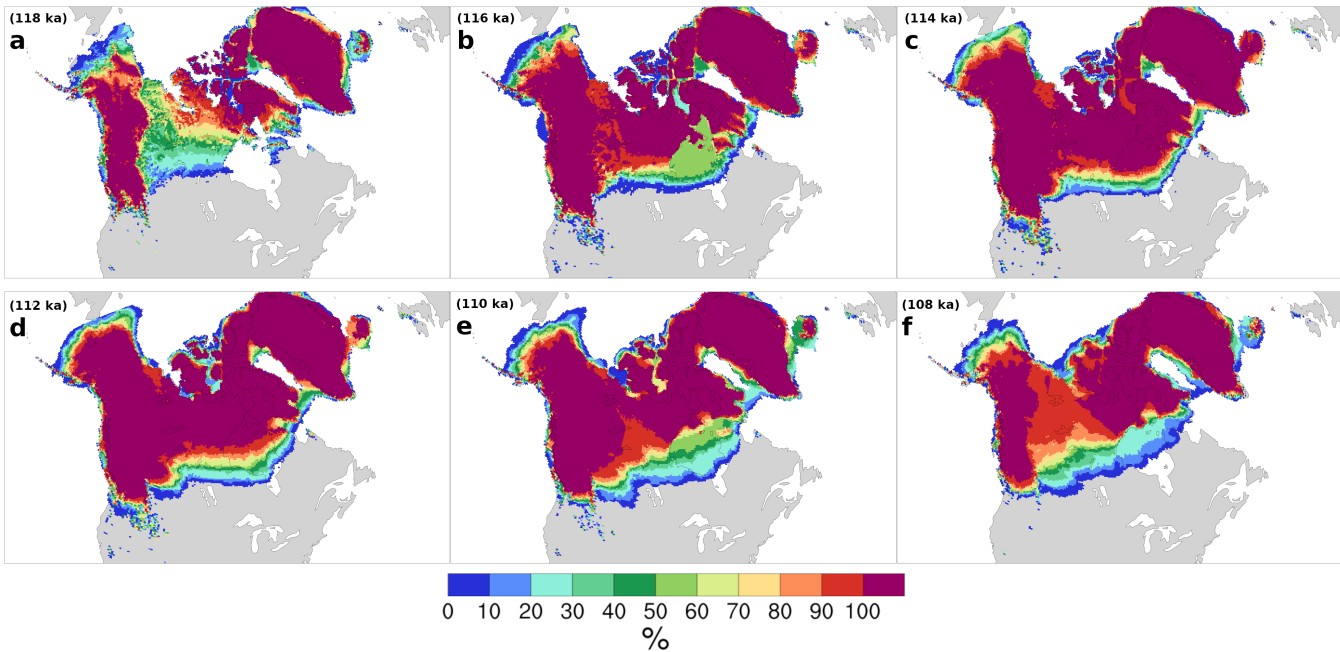

**Figure 6.** NA ice extent ensemble probability distribution at **a** 118 ka, **b** 116 ka, **c** 114 ka, **d** 112 ka, **e** 110 ka, and **f** 108 ka. The 118 ka and 116 ka are included to provide the history before the peak and are not discussed.

are completely ice-free in this region. By 108 ka, the Laurentide and Cordilleran ice sheets are separated in only 10% of the simulations, fewer than 20% of runs simulate a connected Greenland-Iceland ice sheet, and the ice bridge across Davis Strait remains in fewer than 10% of runs.

A key feature from the sample best run snapshots (figure 5) is the continuous slow thickening of Ellesmere Island ice right through to 105 ka. Thus, limited snow accumulation appears to be the major controlling climate factor for this region during LGI. The ice dome north of Hudson Bay also only attains its maximum elevation at 107 ka.

Similar to the early phases of the inception, stadial peak ice extent over EA (116 ka to 112 ka) is more variable between ensemble members compared to NA (figure 7). The maximum continental ice area covered by all runs occurs at 116 ka, with a significant reduction by 114 ka. Fewer than 10% of runs increase their southern ice extent through to 112 ka. Scotland exhibits some ice cover in the majority of runs, but the North Sea remains ice-free.

## 3.2 Temporal pattern of ice evolution across the ensemble

As shown in the previous section, the rates of ice growth and retreat are not consistent through the LGI in all regions, especially in EA. To diagnose the development of these ensemble member differences in time, we subdivide NA and EA into four sectors each (outlined in figure 8) and examine the evolution with time of ice volume in each sector along with correlations between sector maximum ice volumes.

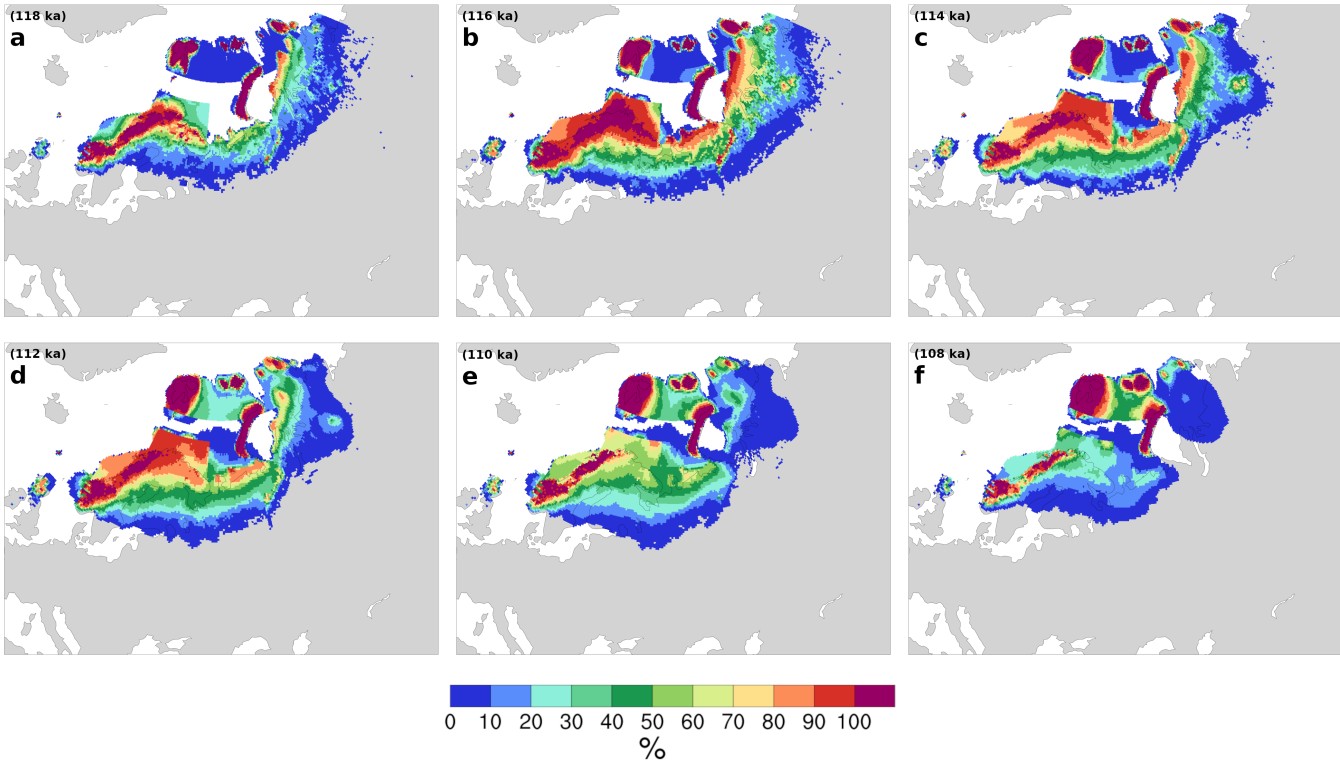

**Figure 7.** EA ice extent ensemble probability distribution at **a** 118 ka, **b** 116 ka, **c** 114 ka, **d** 112 ka, **e** 110 ka, and **f** 108 ka. The 118 ka and 116 ka are included to provide the history before the peak and are not discussed.

The NA sectors include two regions in the Canadian Archipelago ( Ellesmere, $\text{NA}_{El}$, and Baffin Islands, $\text{NA}_{Bf}$), Quebec ($\text{NA}_{Qb}$), and the Rockies ($\text{NA}_{Rc}$). The EA sectors include the north-western Barents Sea and Svalbard ($\text{EA}_{Sv}$), the Kara Sea and nearby land ($\text{EA}_{Kr}$), and eastern and western Fennoscandia ($\text{EA}_{EF}$ and $\text{EA}_{WF}$).

### 3.2.1 North American ice sheet

In all NA regions in figure 9 except $\text{NA}_{El}$, ice volume increases to a maximum sometime between 112 ka and 109 ka and then decreases. In $\text{NA}_{El}$, the coldest region of NA, ice volume increases throughout the LGI in most simulations.

Generally, the ice sheet growth phase for each sector is more consistent between runs than its retreat phase. In sector $\text{NA}_{Bf}$ (figure 9b), ∼10% of simulations lose between 1 and 1.5 m SLE of ice between 112 and 107 ka and maintain a constant ice volume afterwards. The rest of the runs show a range of behaviours, from almost no ice loss to 80% loss. In contrast, in $\text{NA}_{Qb}$, the most southern and warmest sector, the maximum ice volume varies between almost zero to more than 1 m SLE, and no simulation sustains ice cover through to 102 ka. The $\text{NA}_{Rc}$ region spans the widest range of latitudes, but it also contains some of the highest-elevation sites of NA. It shows both strong ice growth and a wide range of ice loss scenarios over the LGI. Notably, ice develops over western NA ($\text{NA}_{Rc}$) at the same time as it is growing in the east.

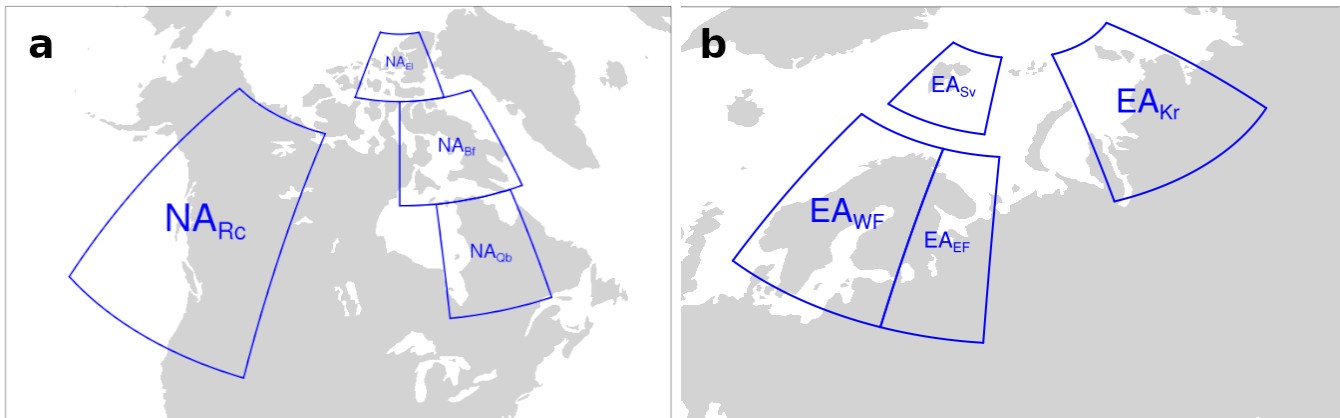

**Figure 8. a**. NA sectors, and **b**. EA sectors

One pattern that emerges most strongly in NA$_{Qb}$ is that the runs with larger ice sheets tend to have delayed peak times. This is consistent with the observation in the previous section that runs with the largest NA ice sheet extent retreat more slowly than those with smaller ice sheets.

### 3.2.2 Eurasian ice sheet

In EA, the most northern (and coldest) sector, EA$_{Sv}$ has steadily increasing ice volume throughout the LGI. This pattern is similar to that observed for NA$_{El}$. Otherwise, the rest of EA sectors show ice growth and retreat patterns similar to NA$_{Qb}$, where there is a wide variation in the total ice volume reached and (near-) complete ice loss by the end of the LGI. These regions also generally reproduce the tendency for larger ice sheets to have later peak ice volumes, ranging between 114 and 110 ka. However, in EA$_{EF}$ and EA$_{WF}$ there are some notable exceptions to this pattern, where some simulations exhibit late peak times (ca 108ka) for a wide range of maximum ice volumes.

### 3.3 Relationships between changes in the North American and Eurasian ice sheets

We have examined the build-up and retreat of ice sheets in NA and EA independently. Past modelling studies indicate that the presence of NA ice can affect conditions over EA (Beghin et al., 2013; Colleoni et al., 2016; Liakka et al., 2016; Ullman et al., 2014; Kageyama and Valdes, 2000) and therefore potentially EA evolution. Thus, we consider next whether there is any evidence for such a relationship acting in this ensemble.

Comparisons of EA maximum ice volume versus NA maximum ice volume in figure 10 indicate that there exists no simple relationship between these two fields. Small NA ice volumes correspond to small EA ice volumes. However, when NA ice volumes are larger, figure 10 suggests a possible bifurcation in the behaviours of the runs: one group of runs exhibits a strong increase in EA maximum ice volume with increasing, but intermediate-sized NA maximum ice volumes. In the second group

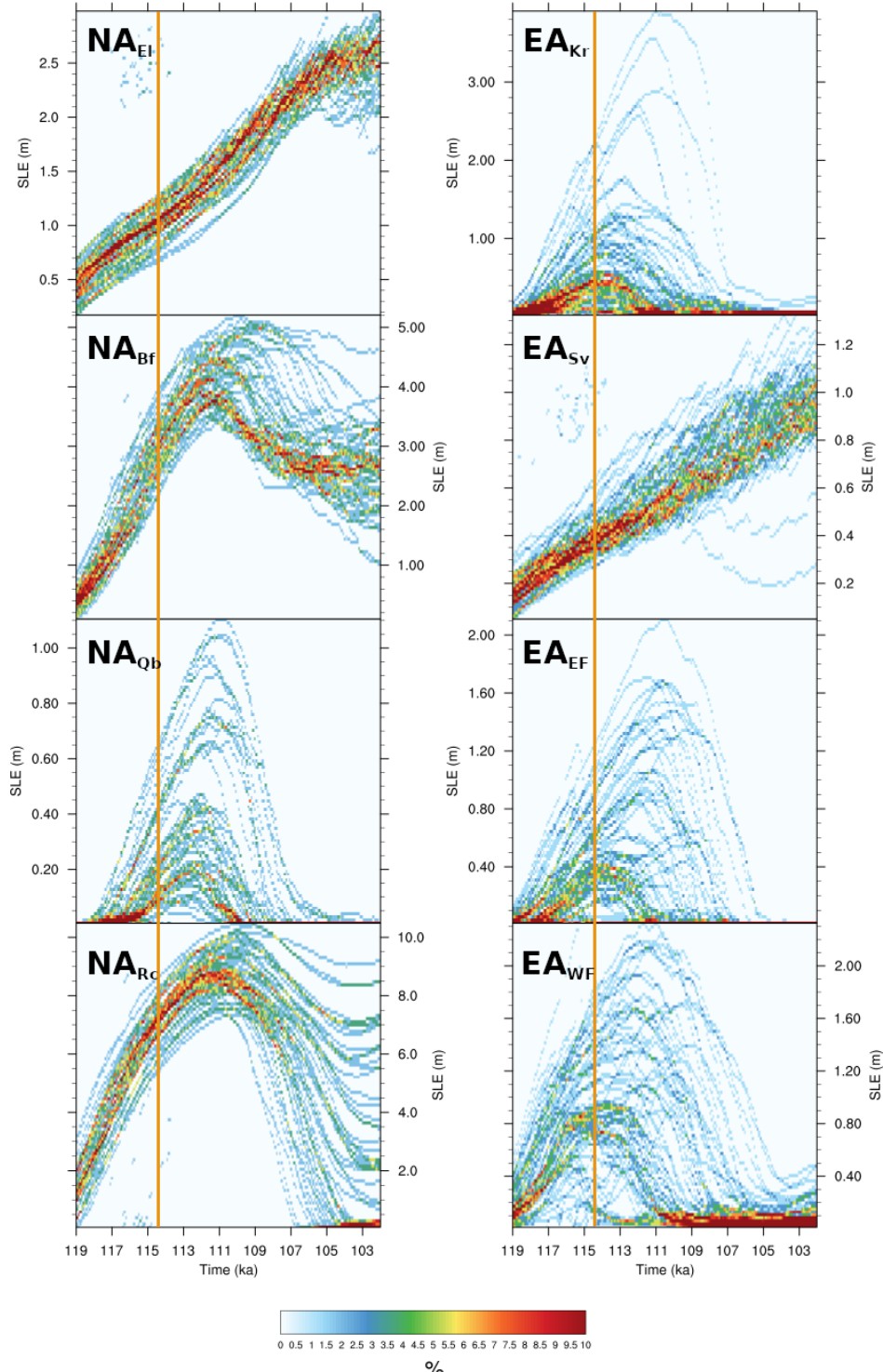

**Figure 9. Left**. NA ensemble distribution of ice volume during LGI in $NA_{El}$, $NA_{Bf}$, $NA_{Qb}$, and $NA_{Rc}$. **Right**. EA ensemble distribution of ice volume during LGI in $EA_{Kr}$, $EA_{Sv}$, $EA_{EF}$, and $EA_{WF}$. The vertical orange line shows the timing of the minimum summer insolation at 60°N.

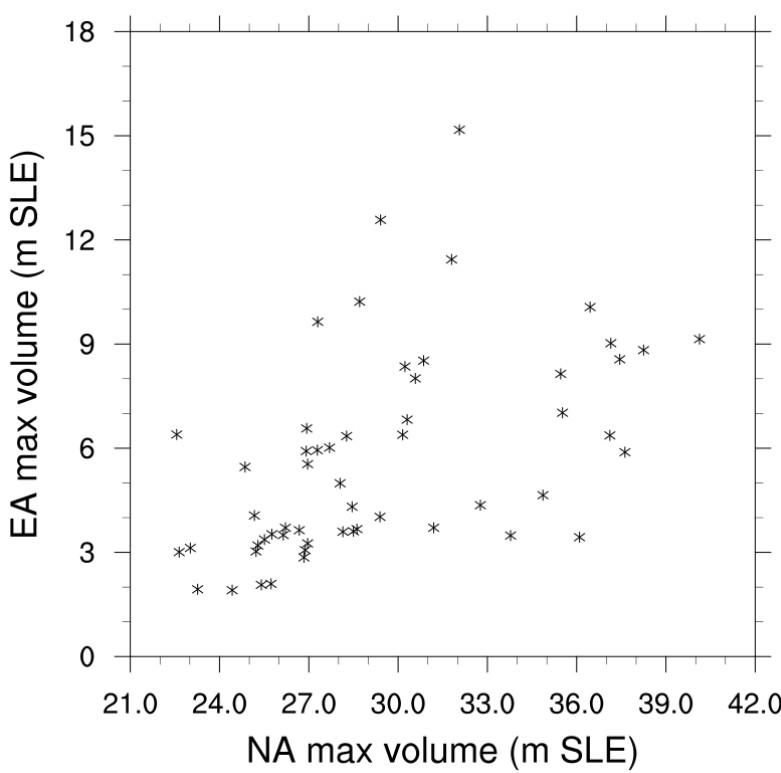

**Figure 10.** The maximum volume of the NA and EA ice sheets for individual runs.

of runs, EA maximum ice volumes remain small until NA maximum ice volumes pass a (very large) threshold. Beyond this threshold, EA ice volumes increase to intermediate sizes with further increases in NA maximum ice volume.

Although there is no simple relationship between the volumes of the NA and EA ice sheets, there is a relationship between
the timing of the peak ice volume for these two ice sheets in most ensemble members. In figure 11a, the peak ice volume and peak ice area nearly always occur earlier in EA than in NA. This result is expected given the smaller size and related stronger sensitivity of the EA ice sheet to orbital forcing. The duration of this lead depends strongly on model parameters and ranges between 200 years to 6 kyr. In a small subset of runs, the EA ice volume peaks early (∼115 ka) or late (∼110 ka) regardless of the timing of the NA ice volume peak (further evidence in support of the aforementioned possible bifurcation).
The correlation in the timing that maximum ice volumes are reached in NA and EA in most runs in figure 11a may indicate that these ice sheets are affecting each other's growth and retreat, or it may indicate that the parameter choices that lead to larger ice sheets in one region also encourage growth in the other. One plausible mechanism whereby the NA ice sheet may affect the development of the EA ice sheet is through a reduction in hemispheric temperatures. However, there is no evidence of this, as the timing of maximum EA ice volume (figure 11b) has no consistent phase relationship with the timing of EA
minimum temperature.

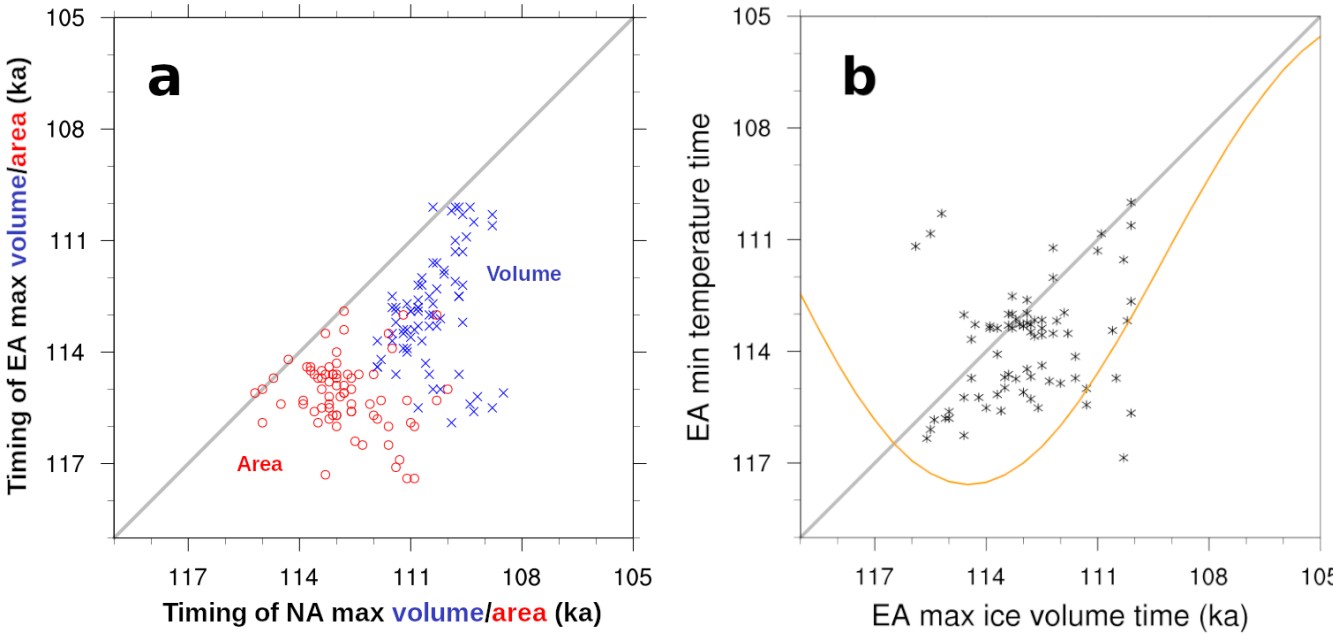

**Figure 11. a**. Timing of the EA ice volume (blue) and area (red) peak with respect to the NA peak time. **b**. Timing of the EA minimum temperature and maximum ice sheet volume for individual runs. The orange curve shows the summer (JJA) insolation at 60°N.

## 3.4 Climate of the Inception

Having documented the trajectory space of ice sheet changes and identified the more robust features in our ensemble of LGI simulations, we now consider relevant controls from the climate system. To that end, we focus on temperature and precipitation as the two main controls on ice sheet thickness and extent (at least for terrestrial components). These are in turn affected by Northern Hemisphere sea ice extent (which alters the exchange of heat and moisture between the atmosphere and ocean), the AMOC (through changes to oceanic heat transport to high latitudes), and the latitude of the jet stream (through changes to atmospheric heat transport and the location of storm tracks).

Northern summertime temperature and annual precipitation are ice-sheet relevant climate characteristics that most directly control ice sheet extent and thickness. For our ensemble, both temperature and precipitation of NA and EA (figure 12) show abrupt reductions early in the LGI interval initially in phase with the reduction in insolation at 60°N. In NA, summer temperature and annual precipitation reach their respective minimum values at 116.8 and 116.1 ka, approximately 2.3 and 1.6 kyr earlier than insolation. An increase in the radiative forcing from changing atmospheric $pCO_2$ (purple time series in 12) after 116.2 ka and especially a subsequent decrease after 114.3 ka approximately correspond with the interval of discrepant NA mean summer temperature change (relative to insolation forcing). Since the relatively high albedo ice sheets and sea ice tend to be fairly extensive by this time (figures 6, 7 and 12), changing insolation will also be a smaller contributor to the regional energy balance. The possible role of changes in AMOC and sea ice cover are examined below.

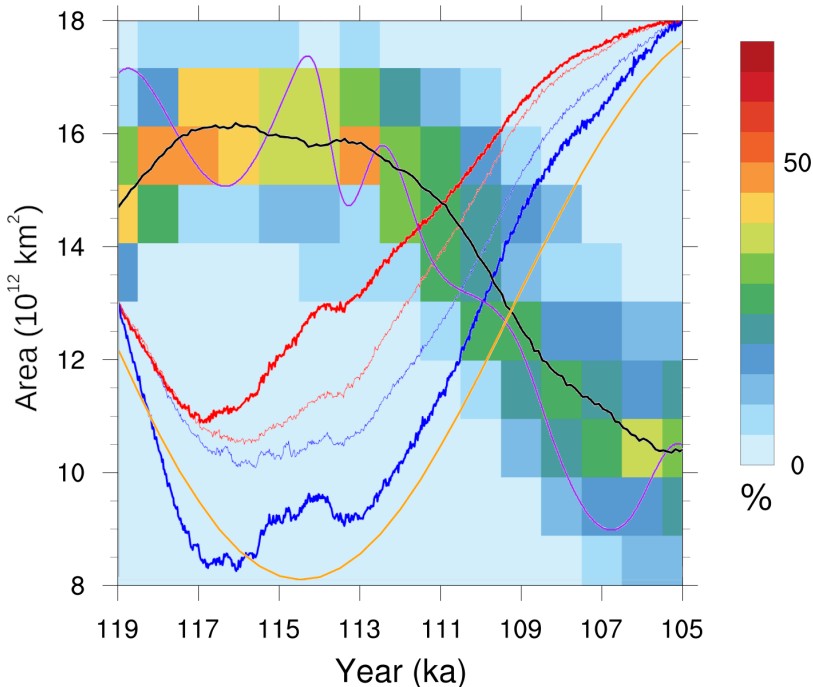

**Figure 12.** The ensemble distribution of Northern Hemisphere late summer sea ice total area. The black line shows the ensemble mean late summer sea ice area. The blue lines show the scaled ensemble mean summer temperature anomaly with respect to 119 ka in NA (thick) and EA (thin). The red lines show the scaled ensemble mean annual precipitation anomaly with respect to 119 ka in NA (thick) and EA (thin). The orange line represents the summer insolation at 60°N. The purple line shows the changes in log(pCO$_2$) to approximately capture its effective radiative forcing. Temperature, precipitation, insolation, and pCO$_2$ are plotted solely for the sake of phase comparison, and therefore their actual values are not indicated.

The early stages of ice growth in NA appear to be dominated by ice sheet expansion in response to regional cooling, since precipitation is decreasing. In 119 to 117 ka snapshots of near-surface temperature and ice sheet elevation from a single simulation in figure 5, the southern ice sheet margins tend to be located between the -2°C to 0°C JJA isotherms in most regions except for those with high levels of accumulation (*e.g.* the Rockies). NA ice sheet area reaches its maximum after the temperature and precipitation minima, between 114 and 113 ka (see figures 11a and 12). Thus, both temperature and precipitation are increasing at the time that the maximum NA ice sheet area is reached.

NA ice volume continues to grow until approximately 111 ka through a thickening of the ice sheet (*cf.* figure 5) in response to increasing precipitation under continuing cold temperatures. During the 113 to 111 ka interval, the low elevation sectors of the southern NA ice margin in the sample simulation in figure 5 is generally between the 4°C to 2°C JJA isotherms. Eventually, the NA ice sheet begins to lose mass after further increases in temperature and precipitation. At this time, the southern margin of the ice sheet tend to fall south of the 4°C isotherm.

In EA, temperature and precipitation also show an abrupt but weaker reduction in the early inception. The ensemble mean EA summer temperature and precipitation minima have a longer duration interval than that of NA and show little sensitivity to the $pCO_2$ changes. The reasons for this result are as yet unclear. The onset of renewed EA warming and increasing precipitation correspond to the maximum extent of EA ice sheets. However, EA ice volume continues to grow for as much as another 3 kyr. EA temperature and precipitation gradually increase until ∼113 ka, when the sea ice area starts to decline (figure 12). After this time, both temperature and precipitation increases accelerate.

The southern margin of the EA ice sheet largely mirrors the relationship to surface isotherms in NA, at least for the sample run in figure 5. During the growth phase, the ice margin tends to lie in between the 2°C to -2°C JJA isotherms. By 109 ka, the EA southern margins are generally south of the 4°C isotherm.

In assessing the contributions of sea ice, the AMOC and the jet stream, summer sea ice has the strongest correlations with temperature and precipitation changes in EA. Late winter sea ice area shows no consistent pattern of change over this time period and is not related to ice sheet volumes in either NA or EA (see supplemental figure 1). However, its summer extent varies in correspondence with Northern Hemisphere temperatures: it peaks prior to the minimum in insolation at 60°N, remains extended, and then decreases. The onset of major sea ice retreat at approximately 113 ka is in phase with a rapid acceleration of both NA and EA summer warming and annual-mean precipitation. Deciphering the causal relationships of this phasing requires future sensitivity experiments. However, one can infer that sea ice likely has a positive feedback role for both precipitation and temperature at this time.

Neither the AMOC (nor meridional heat transport in supplemental figure 3) nor the wintertime jet stream exhibit any clear, consistent changes that coincide with temperature and precipitation or ice sheet changes. In 80% of the runs, the AMOC gradually increases during the glacial inception to a maximum of 22 Sv around 108 ka (not shown). After this, it decreases once more to its initial values of 16 to 18Sv. In the remaining 20% of runs, the AMOC oscillates within its initial range of values.

Similarly, the minimum latitude of the North Atlantic mean winter jet stream is restricted to 43 to 47°N with the only significant change over time being an increase in the fraction of runs with the more southern position (with greater than 70% of runs by 104 ka). Previous work indicates that the latitudinal position of the winter-time North Atlantic jet stream depends on the latitude of the south-eastern margin of the NA ice sheet (Andres and Tarasov, 2019). For the current ensemble, the NA ice sheet remains north of the preferred latitude for the jet stream at all times, so the ice sheet is unable to directly influence the jet stream in this way.

However, the minimum latitude of the summertime North Atlantic jet stream does vary in concert with NA ice sheet and sea ice extents. Specifically, the most commonly occupied position of the summertime jet changes from 52°N to 48°N during the 117 to 116 ka interval (figure 13), in correspondence with significant initial ice growth over the $NA_{Qb}$ sector (the most proximal sector diagnostic for central and eastern Canada in figure 9 ) . The subsequent northward migration occurs across the ensemble from 110 to 107 ka, again in correspondence with the wider cross-ensemble range of deglaciation times for the $NA_{Qb}$ sector. The much warmer JJA temperature during 107 ka compared to 119 ka in figure 1 likely explains the higher latitudinal

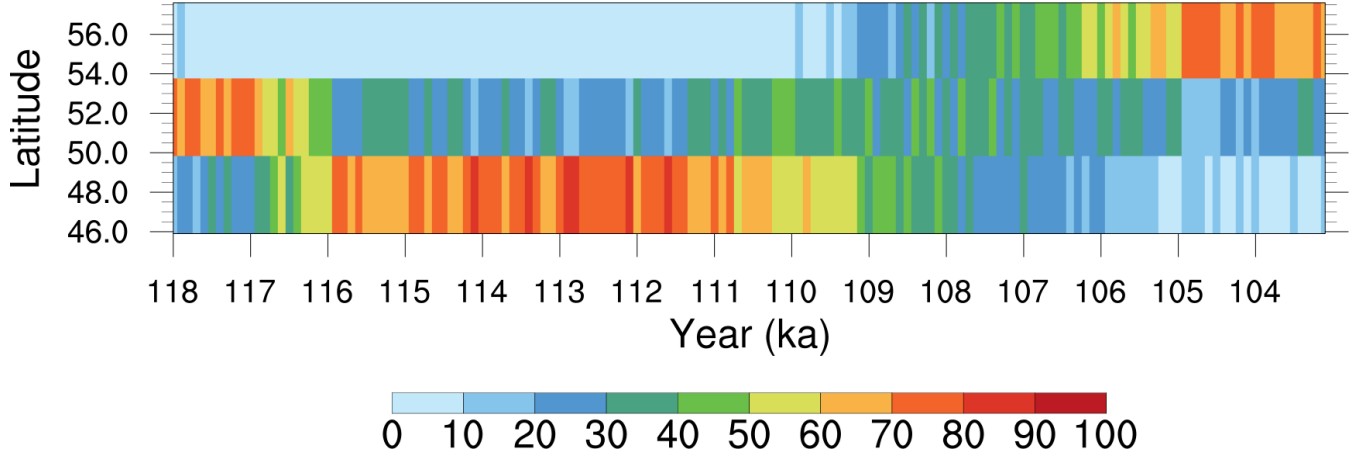

**Figure 13.** Ensemble distribution of mean most southern latitudinal position of the JJA jet of the Atlantic Ocean.

position (56°N) of the ensemble mean summer jet at 107 ka compared to that of 118 ka. These shifts in the jet stream likely affect summertime temperature and precipitation over EA.

## 4 Discussion

### 4.1 Present day temperature bias

In interpreting ensemble results for LGI, it is important to be cognizant of present-day model biases. However, we add a caveat that the applicability of present-day biases to LGI conditions, especially once there is extensive ice cover, depends on the source of the biases. If the biases arise due to a mis- or under- represented stationary wave response to present-day topography in LOVECLIM (e.g. due to its low spatial resolution, Lofverstrom and Liakka (2018)), then the magnitudes and locations of these biases are expected to change when the stationary waves change in response to the growth of continental ice sheets.

Figure 14 shows the present-day mean June/July/August temperature bias of the 55-member ensemble for both NA and EA. Model temperature biases under present-day conditions are larger over NA than over EA and the $9.5^oC$ to $14^oC$ mean regional summer temperate biases would have hindered at least initial glaciation and likely LGI maximum extent (to a more uncertain degree) over Hudson Bay, Quebec, and Labrador. However, though the ensemble mean summer temperature bias is stronger over Hudson Bay than the adjacent sector of Quebec/Labrador, this doesn't preclude complete glaciation over Hudson Bay (but not James Bay) by 112 ka (figure 6) in ensemble members. Increased LGI stadial ice over these latter regions would also improve fits to global mean sea-level proxies (*cf.* Figure 1). Thicker stadial ice could also enable a stronger and faster post-stadial retreat.

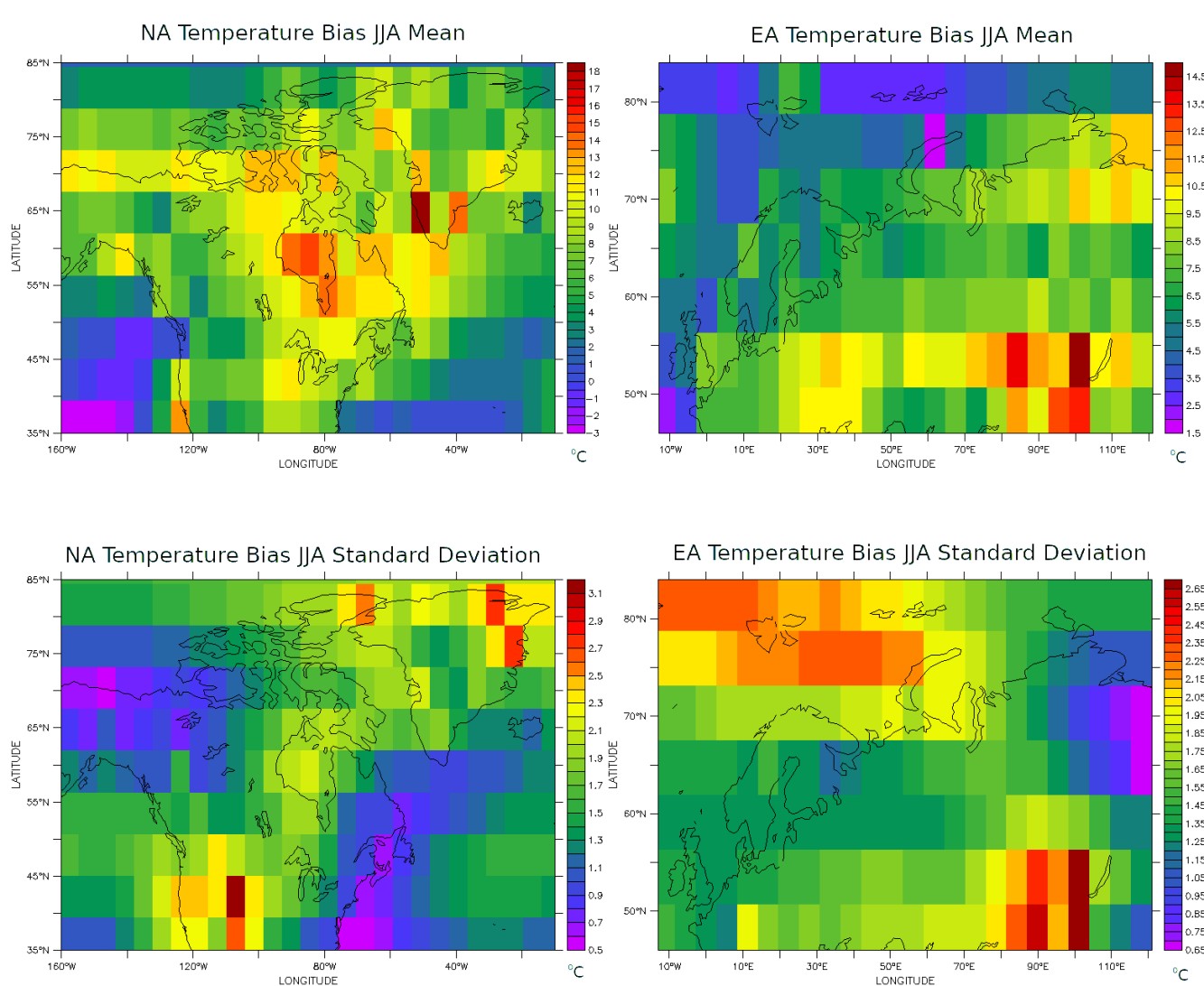

**Figure 14.** Mean present-day summer (June/July/August) temperature bias of the reduced 55 member sub-ensemble relative to the NCEP reanalysis climatology (Kalnay et al., 1996). Climatological temperatures from LOVECLIM simulations were adjusted for elevation differences between the two datasets using the LOVECLIM- derived vertical surface temperature lapse rate for each ensemble member.

## 4.2 Caveat about marine sectors

Ice sheet growth in marine sectors is found to be highly sensitive to the treatment of sub-shelf melt, even at high latitudes. This is particularly evident in figure 7, where marine ice sheet margins are at times extended straight lines. These lines match the boundaries for different ocean temperature sectors in LCice, which propagates the vertical temperature profile from assigned upstream diagnostic sites to whole downstream ocean sectors for computing submarine ice melt (Bahadory and Tarasov, 2018). This artifact of the model setup underlines the potentially important role of ocean temperatures on submarine melt and its control of marine ice extent.

The crude shallow ice approximation treatment of ice shelves in the utilized version of the GSM along with the continuing challenge for the community to find a well constrained ice calving representation are further contributors to uncertainty in the marine sector results of the model. The GSM has been recently revised with the inclusion of shallow shelf approximation ice dynamics and ongoing work will examine the impact of this and other model updates on resultant modelled LGI ice evolution.

## 4.3 Widespread snowfield glaciation versus spreading from high elevation nucleation sites

Our results provide a sensible merger of the two contrasting hypotheses. Glaciation starts with nucleation over high latitude and high elevation regions, but widespread snowfield thickening subsequently creates thin ice ($< 500$ masl) over expanses of continental northern sectors for both NA (by 118 ka) and EA (between 118 to 117 ka). This is clearly visible for our sample best fit run (figure 5).

## 4.4 The challenge of excessive Alaskan glaciation

The one significant transgression of inferred Late Pleistocene glacial limits in our ensemble is near complete glaciation of Alaska (figure 6). This is contrary to geological inferences (Kaufman et al., 2011). Unless these inferences are incorrect, then the approximately 2 to 4 m SLE contribution to the inception peak from glaciation of central Alaska in our ensemble should be removed from our ensemble total.

Excessive glaciation of Alaska is a common problem for models (*e.g.* Bonelli et al., 2009). Past studies indicate at least two factors may resolve this problem: atmospheric model resolution (and/or complexity) and changes in snow albedo due to dust deposition. Though still displaying somewhat excessive Alaskan ice coverage, Herrington and Poulsen (2011) avoid complete glaciation with fixed 116 ka boundary conditions using the GENESIS AGCM (and slab ocean) at T31 resolution. A glacial decrease in surface air pressure over the Bering Strait region is apparently associated with an increase in northward transport of sensible heat towards Alaska. Whether this suppression of Alaskan glaciation is solely due to increased atmospheric model resolution or complexity is unclear. It is also unclear if the result Herrington and Poulsen (2011) would persist with a fully coupled ocean model.

Using the CLIMBER EMIC, Ganopolski et al. (2010) obtain reduced though still excessive Alaskan glaciation. A previous study traced much of this reduction to the inclusion of aeolian dust forcing on snow albedo for the surface mass balance determination (Calov et al., 2005b). However, confidence in these results is limited given the crude determination of dust

deposition and associated albedo changes in their model. More advanced studies have verified the significant impact of dust on snow albedo (Krinner et al., 2006, though with an imposed dust-deposition rate) but have also found it difficult to obtain even the magnitude of dust deposition (Mahowald et al., 2006) inferred from extensive loess deposits in Alaska (Muhs et al., 2003). A potentially critical role for changing dust deposition in suppressing Alaskan glaciation is therefore plausible, but in need of more advanced modelling.

The CAM3 atmospheric general circulation model (at T85 spectral resolution) produces warmer and drier conditions over Alaska under Last Glacial Maximum (LGM) boundary conditions due to two processes: 1) a reduction in local cloudiness due to the combined effects of colder sea surface temperatures and descending air from the topographic high pressure system nearby, and 2) a southward shifting of North Pacific storm tracks away from this area (Lofverstrom and Liakka, 2016). The degree to which LCIce is able to resolve these two phenomena is not clear. Present-day LCIce warm summer temperature (figure 14) and high annual precipitation biases (supplemental figure 5) over Alaska would correspondingly hinder and facilitate regional glaciation.

## 4.5 Brief comparison to past geological inferences

Aside from global sea level constraints and maximal extent bounds from subsequent marine isotope stage (MIS) records, there is very little known about NA ice extent during LGI. Stokes et al. (2012a) provides the most recent review of geological inferences and modelling results for NA LGI. The main discrepancies in our results are the already noted issue of excessive Alaskan glaciation and likely inadequate ice extent over Quebec and Labrador.

Batchelor et al. (2019) provides the most recent synthesis of geological inferences for Northern Hemispheric ice sheets through the Quaternary. Most telling of the limited constraint for NA LGI is their supplemental figure 6 and associated table 9 which lists only one empirical timeslice reconstruction (and no empirical data points) constraining their minimum and best guess maximum NA MIS 5d (108-117 ka) ice extent and as well as the lack of empirical constraints for MIS 5c (92-108 ka). The empirical reconstruction source (Kleman et al., 2010) has a large age uncertainty and favours MIS 5b (86-92 ka) for this timeslice. However they could not rule out that the flow sets used for the timeslice reconstruction were from a pre-Eemian stage. Their complete lack of MIS5d glaciation of Hudson Bay contradicts the favoured inference of Clark et al. (1993), pointing to the challenge of inferences from sparse geological data with very poor age control. It should be noted however that Batchelor et al. (2019) barely reach the inferred MIS5d sealevel minimum of Spratt and Lisiecki (2016), using an ice volume to area scaling relationship derived for a circular mono-dome ice sheet with plastic rheology (Cuffey and Paterson, 2010). We find that this relationship over-estimates ice volume during last glacial inception by at least 50% when compared to the base GLAC1-D nn9927 ice sheet chronology from the calibrated glaciological modelling of Tarasov et al. (2012).

Eurasia as a whole also lacks a clear geologically-inferred LGI stadial extent. Only in the Fennoscandian sector are there published geologically-inferred LGI stadial extents (Batchelor et al., 2019). However, the geologically-inferred Early Weichselian (MIS 5) ice extent maximum of Svendsen et al. (2004, nominal 90 ka in) generally encloses (and for much of the southern margin largely tracks) the 50% ensemble distribution (figure 7). The main regional exceptions are more extensive ice on the western coast of Svalbard and extensive marine ice on the western Norwegian coast.

We leave this subsection purposely brief in the hope that members of the geological community will execute detailed and up-to-date comparisons with our ensemble chronologies. A first example of such is the review for this paper provided by John Andrews (Andrews, 2020) which provides a useful discussion of existing possible paleo constraints on four features of this ensemble (including the Denmark and Davis Strait ice bridges and the relatively symmetric rates of glaciation and subsequent retreat).

## 4.6   Is there a single very likely spatio-temporal pattern of LGI ice sheet evolution?

    To partially characterize the range of the spatio-temporal patterns of ice sheet evolution in our ensemble, we consider the intersectorial relationships of maximum ice volume for each ensemble run (figure 15). The absence of correlation in maximum ice volumes for different sectors will indicate that that there are multiple temporal patterns of ice development in these regions.

    For NA, the northern Arctic (NA$_{El}$) sector maximum ice volume has no obvious correlation with that of other sectors. This
is consistent with the continual growth of ice throughout the simulations in this region. All other NA sectors display relatively strong correlations aside from a threshold response for NA$_{Qb}$ relative to the Pacific Cordillera (NA$_{Rc}$).

    For EA, again the most northern and continuously growing sector (EA$_{Sv}$) has relatively no correlation in maximum ice volume with other sectors (figure 15). The relatively northern and largely marine eastern sector (EA$_{Kr}$) has a strong correlation with the two Fennoscandian sectors for the 5 runs with maximum (EA$_{Kr}$) greater than 2.5 mSLE. For the other runs,
the correlation is much weaker and with a much lower mean slope, perhaps indicative of a threshold in ocean temperatures controlling subshelf melt and enabling ice calving.

    There are no strong correlations between NA and EA regions (c.f. supplemental figure 4). There is moderate correlation between the Baffin Island sector NA$_{Bf}$ and western Fennoscandia EA$_{WF}$, perhaps reflecting ocean circulation connections between Baffin Bay and the GIN Seas. More limited correlations exist between NA$_{Bf}$ and eastern Fennoscandia (EA$_{EF}$) and
between the western Cordillera (NA$_{Rc}$) and western Fennoscandia (EA$_{WF}$). The only other possible relation of note is the absence of large maximum ice volumes for the eastern Kara Sea region (EA$_{Kr}$) when ice volumes are near maximal for all NA sectors south of Ellesmere (with only 5 runs for this case, the relationship is tentative).

    The only clear indication of a bifurcation in regional temporal evolution is the presence of both early and late timing of maximum regional ice volume for Fennoscandia (EA$_{EF}$ and EA$_{WF}$) for a range of regional maximum ice volumes (figure
9). The extent to which possible associated bifurcations in sea ice extent and stationary atmospheric waves (described in the results section) may play a role in this must await future analysis.

## 4.7   Equilibrium climate response

    The equilibrium warming of all 500 ensemble members is between 1.1 and 2.3°C in figure 16. The full ensemble therefore brackets the lower bound (1.5°C ) but not the upper bound (4.5°C ) of the IPCC AR5 likely range for ECS (Flato et al., 2013).
The constraint of capturing "acceptable" LGI growth and retreat (*i.e.* the "successful" 55 member sub-ensemble) increases the lower bound to 1.4°C, still below that of the IPCC AR5 likely range, but does not affect the upper bound.

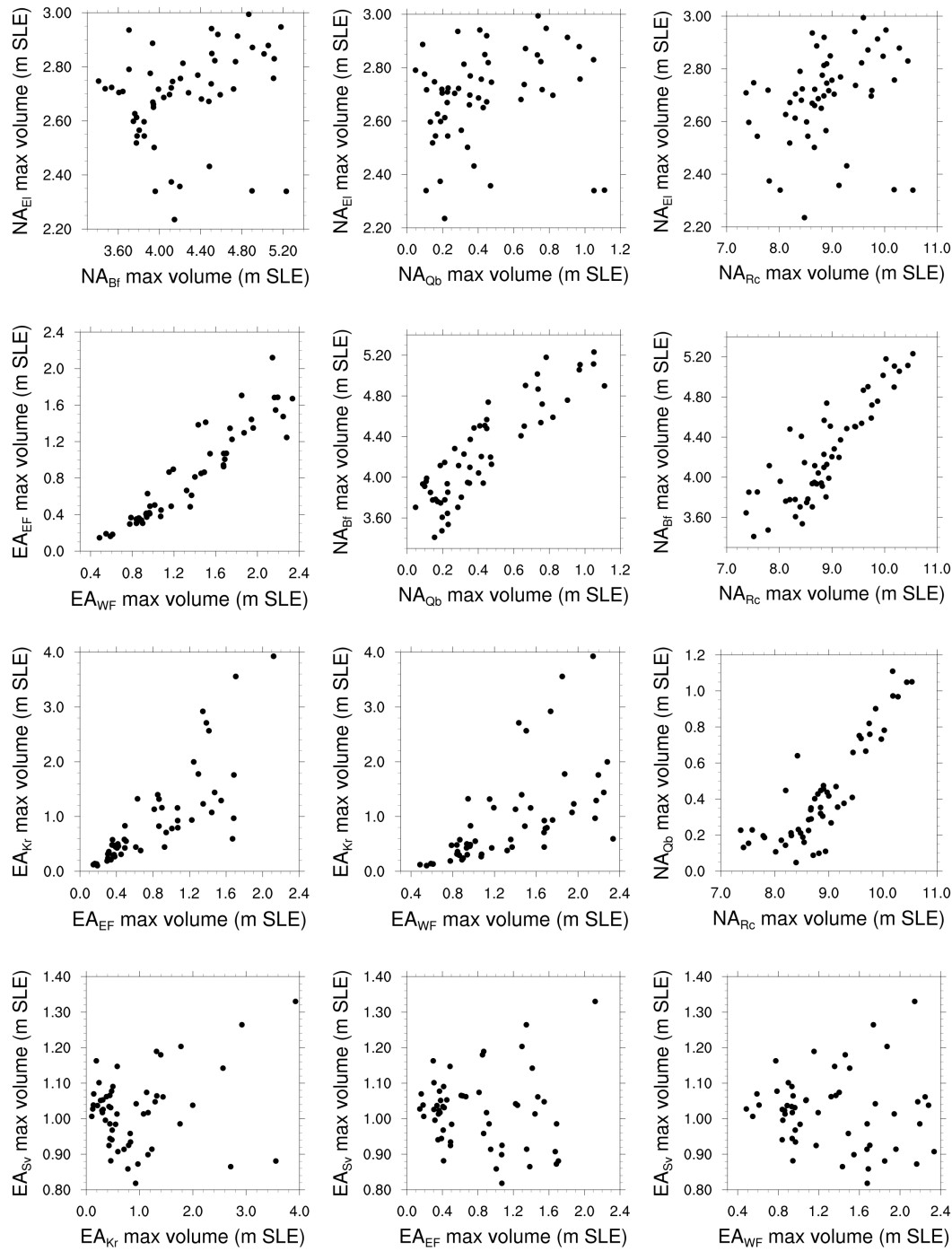

**Figure 15.** Correlation plots of maximum ice volume for NA and EA diagnostic sectors (figure 8).

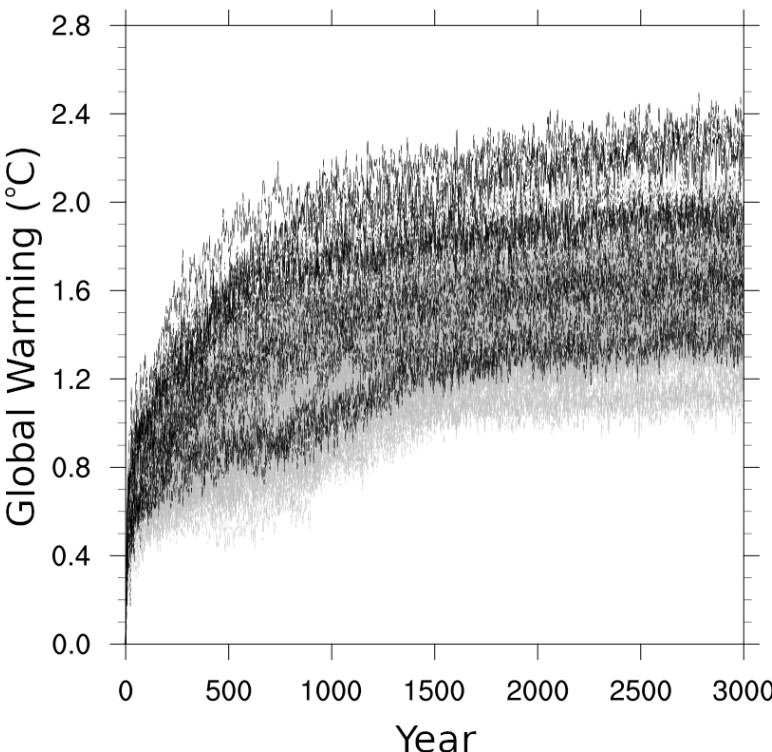

**Figure 16.** The mean global equilibrium warming projected by the selected 55 ensemble members (black lines) and the 500 inception simulations (grey lines) in the ECS experiment (refer to the text for details).

Given the simplified physics, limited climate model resolution of LCice, and the tendency of LOVECLIM itself to produce low ECS values (Flato et al., 2013), the actual ECS ranges derived here have limited credible value in constraining future climate change. However, it may also be that a stronger threshold for what is "acceptable" may further increase the LCice lower bound for ECS. Clearer constraints on LGI sea level history would be of value here. To acquire any significant confidence, this increased lower bound requires replication by more advanced models. However, as is, the result suggests that LGI replication in coupled ice/climate modelling has potential value for constraining climate model sensitivity and therefore constraining future climate change.

## 5  Conclusions

We used LCice 1.0, a two-way coupled ice sheet and climate model, to generate an ensemble of 500 transient simulations of the LGI that differ according to the combination of parameters and parameterizations used in the climate component (LOVE-CLIM), the ice sheet component (the GSM) and the coupling between them. Of these 500 simulations, 55 simulations passed our ice volume evolution acceptance criteria for the LGI. In this paper, we document the patterns of ice growth and retreat exhibited by North American and Eurasian ice sheets in these 55 runs.

We applied two tests of the representativeness of these simulations to historical changes during the LGI: comparisons of total sea level changes with time, and comparisons of near-surface air temperatures at the location of the GRIP ice core. Maximum LGI ice volume is under-estimated in the ensemble relative to that inferred by Lisiecki and Raymo (2005), although it lies within the collective uncertainties of the three proxy reconstructions considered herein. The timing of the LGI sea level minimum in our modelled ensemble does not match any of the reconstructions considered here, but it is bounded by them: it

occurs approximately 2 kyr earlier than that of the Lisiecki and Raymo (2005) and Waelbroeck et al. (2002) reconstructions but less than 1 kyr after the sea level minimum in Siddall et al. (2003). Discrepancies are likely due to the absence of a modelled (and probably out of phase) Antarctic ice sheet contribution in LCice 1.0, model limitations (as evidenced by the present-day warm summer temperature bias over Hudson Bay and Quebec), and dating uncertainties in the proxy-based reconstructions

The ensemble-mean temperature is in approximate agreement with an inverse reconstruction from the GRIP ice core during

the LGI cooling phase. Subsequently, a strong warming in the model driven by orbital and greenhouse gas forcing is absent in the reconstruction. Given regional warming is robust across the ensemble and the lack of a plausible physical mechanism to sustain cold, stadial conditions under increasing insolation, we suggest the discrepancy may be due in part to uncertainties in the $\delta^{18}$O to temperature inversion. This may also explain in part why the model also fails to capture the millennial scale variance of the proxy record.

The regional LGI pattern of initial ice growth and evolution in NA and EA is consistent with the high elevation and high latitude nucleation paradigm (first over Ellesmere, Svalbard and Franz Joseph islands, then the northern Rockies, and Baffin and Novaya Zemlya islands). Subsequent nucleation over lower latitudes is followed by large-scale snowfield expansion/thickening over central northern Canada, merging eastern and western NA ice in all runs by the time they reach their maximum LGI ice sheet area. EA ice areas and volumes reach their LGI maxima prior to NA in nearly all runs. The maximum ice area for both

NA and EA tends to be reached 2-3 kyr earlier than its corresponding volume.

The EA ice sheet is more sensitive to orbital forcing and ensemble parameter values. At its maximum area, it varies between a single, consolidated ice sheet to multiple, isolated ice caps. After the LGI total ice volume maxima, retreat happens across most sectors except for continued (though slower) growth in the most northern Ellesmere and Svalbard sectors. Aside from the latter, EA tends to have almost complete ice loss by 104 ka.

The southern margin of both ice sheets generally progress from falling between the 2°C to -2°C JJA isotherms during the growth phase to a location south of the 4°C during the peak retreat phase. This progression to warmer isotherms is due to a combination of increasing precipitation and enhanced ice flux to the southern margin (given the thicker upstream ice during the retreat phase). The post-LGI stadial ice mass loss rate and temperature and precipitation increases in EA have higher correlation with sea ice retreat compared to that for NA ice, temperature, or precipitation.

Two perhaps novel features pertaining to NA and Greenland may be of interest to glacial geologists and paleoceanographers. The Greenland ice sheet and Icelandic ice cap are connected in all runs by 114 ka. Furthermore, there is an ice bridge between NA and Greenland across Davis Strait in approximately 80% of ensemble runs. These results have low confidence given limitations in the marine sector of the current version of LCice. Ongoing work with an improved version of LCice will provide a more confident assessment of the plausibility of these two features.

Finally, we assessed how the imposition of minimal capture of proxy-inferred sea level changes during the LGI as an ensemble sieve affected the distribution of equilbrium climate responses (in the coupled model) to a doubling of atmospheric pCO$_2$. For our LCice 1.0 ensemble, we find a 0.3°C increase on the model lower bound when the capture of LGI is imposed.

As an initial attempt with a highly nontrival modelling system, this study has much room for improvement. Ongoing work includes using a significantly revised version of the GSM that includes hybrid shallow shelf/ shallow ice dynamics, explicit insolation dependence of the surface mass-balance computation, a much larger dimension of LOVECLIM ensemble parameters, and inclusion of the Antarctic ice sheet. We are also examining options for climate model bias correction that do not assume present-day biases remain unchanged through a glacial cycle.

An intended contribution of this study is to foster new research about the LGI. We are making a high variance subset of the simulations described in this paper publicly available via an online archive for other groups to use. We especially hope that the field data community will use this archive to test, refute, and/or validate which, if any, of the model-derived LGI trajectories (and characteristics thereof) are consistent with the paleo record.

*Data availability.* Temperature, precipitation and ice thickness fields fields from a high variance subset of ensemble runs has been permanently archived on http://doi.org/10.5281/zenodo.4035034.

*Author contributions.* Taimaz Bahadory ran the model ensemble, carried out the ensemble analysis, and wrote the initial draft of this submission. Lev Tarasov conceived the project, co-designed the experimental/analysis plan, and provided extensive editorial contributions. Heather Andres also provided extensive editorial contributions.

*Competing interests.* The authors have no competing interests.

*Acknowledgements.* The authors thank Marilena Geng for editorial help. This work was supported by a NSERC Discovery Grant (LT), the Canadian Foundation for Innovation (LT), the Atlantic Computational Excellence Network (ACEnet), the Canada Research Chairs program (LT), the CREATE training program in Climate Science (LT), and by the German Federal Ministry of Education and Research (BMBF) as a Research for Sustainability initiative (FONA) through the project PalMod. We thank John Andrews and Andrey Ganopolski for thoughtful reviews.

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
