# Peer review of "Last glacial inception trajectories for the Northern Hemisphere from coupled ice and climate modelling"

_Climate of the Past, 2020_

## Referee Comment (RC1) · John Andrews (Referee) · 30 Mar 2020

**Review: Climate of the Past**

It is 44 yrs since Molly Mahaffy and I attempted the first model simulation to test how to rapidly grow the LIS and mimic the sea-level responses during MIS5. It is also 44 yrs since, working with Dick Peltier, an effort was made to combine an integrated approach to the world's ice sheets volumes, retreat, related changes in sea level, and glacial isostatic adjustment (ICE-1). The present paper by Bahadory et al shows how far the community has come in tackling this problem. My discussion is more along the lines of a comment rather than a review, although my response is also limited by the

fact that, like many, I am not able to access my office and references. The models used in the paper grow the ice sheets vary the climate parameters—the check on the results is the estimated change in sea level. But for many readers the important point that this paper make is in section 4.4 "Brief comparison to past geological inferences" —it is indeed brief a mere 8 lines but this statement outlines the other important, indeed critical, verification of the modelled ice sheets and their expansion and retraction, that is the aerial extent of the ice sheets, a necessary but not sufficient parameter in the calculations of ice volume an global sea level. It is a call for action to the glacial geological community, however, the problems have not changed significantly since the Clark et al 1993 paper-that is the ability to provide a date on buried stratigraphic units, primarily tills, that are older than the  $\sim$ 50,000 radiocarbon dating limit—this problem remains. Generally, the Early Wisconsin Glaciation is correlated with MIS4. Figure 4 is important as it shows the sea level estimates 119-105 ka, but I had a hard job distinguishing the "purple" colors: critically these estimates varying by 20 m, or equal to several Greenland Ice Sheets. It is, however, important to note that these reconstructions do not include the MIS 5d history of the Antarctic ice sheet. I will comment on four conditions that emerge from the model studies in this paper and that I have some knowledge about: namely 1) ice coverage across Denmark Strait between Iceland and East Greenland; 2) the ice bridge between West Greenland and Baffin Island, i.e. across Davis Strait; 3) the fact (their Fig. 5) that the modelled growth indicates that a large glacial lake would be dammed in southern Hudson Bay by 116 ka; and 4) the rate of ice growth versus retreat. Core MD99-2323 is at 1 km depth on the Snorri Drift just south of Denmark Strait—the core extends into MIS7 (Dunhill, 2005, PhD Univ. Colorado). The bedrock of both Iceland and East Greenland is predominantly basalt with high weight % of plagioclase and pyroxene and no guartz (Andrews and Vogt, 2020). If the strait was covered by an ice shelf/grounded ice, this would limit the export of sediment through Denmark Strait and specifically would curtail the export of guartz. A significant peak in guartz occurs at ~106 ka (Andrews and McCave, in prep) but this is relatively minor compared to large peaks at 80 and 30-40 ka. Many glacial
studies indicate that the LGM was more extensive than during the LIG so these data raise the question of what would the modelled ice extent look like during MIS4 and MIS2? It is not clear whether the ice bridge across Davis Strait was grounded or not, if it was that would mean a huge lake behind the damn and a catastrophic flood event when it was finally broken. Along the eastern coast of Baffin Island are several complex cliff sections that record the advance and recession of ice and dated as early as the late Pliocene (Refsnider et al., 2013). A re-assement of the glacial succession using a combined amino acid racemization and cosmogenic radionuclides resulted in units previously assigned to MSI 5 and MIS 5/4 to be allotted ages > 130 cal ka BP. U-series dates (Szabo et al., 1981) on marine molluscs indicated a minimum age of  $\sim$ 70 ka for the Kogalu member compared to  $160 \pm 140$  ka based on the amino acid/radionuclide calculations (Refsneider et al., 2013). The main point here being to stress the considerable difficulties in verifying any MIS 5d glacial configuration solely on the basis of land-based stratigraphic sequences. A solution may exist in the form of marine cores. In Baffin Bay, core HU200829016 extends back to an estimated 120 ka (Simon et al., 2012). The basal unit (MIS 5d?) is a detrital carbonate-rich unit that represents glacial erosion of Paleozoic carbonate bedrock laying on and in the Canadian Arctic Islands and Channels. This would support that the 116 ka scenario (Fig.5) but it is worth noting that calls for an LGM ice shelf in Baffin Bay are not supported by marine evidence (Jennings et al., 2019). No fine-grained (lake sediments) were noted in the core logs (Simon et al., 2014). The MIS 5d reconstructions (Fig. 5) show ice extending NE-SW across Keewatin to Ungava and across Hudson Strait, a consequence of which is that a lake is dammed in Hudson Bay, as was speculated by Adam (1976). The terrestrial record of the last interglacial south of Hudson Bay, the Missinaibi Formation, records a succession from isostatic recovery to glacial inception (Skinner, 1973). On Adam Creek, for example, Missinaibi glaciolacustrinerhythmites underlie early Wisconsinan Adam Till, which might be used to validate the situation in Fig. 5, although a) it is not known whether the glaciolacustrine sediments are extensive, and b) Adam Till indicates a flow from the Quebec/Labrador dome (Thorleifson et al., 1992), which is

**CPD**
contrary to the inferred ice build-up. However, the ages of the Wisconsinan sequence in Hudson Bay remains uncertain as shown by the discussion between Dalton et al (2019) and Miller and Andrews (2019). The authors note that contrary to the general pattern of sea level change (saw-toothed curve) this is not apparent in the MID 5d sea level reconstructions. It is far from clear why this should be the case as to have ice sheets advancing into fiords, straits, and marine embayments should be faced with the same issues that we see happening today, i.e. rapid calving, so it is difficult to see why the growth and decay should be more symmetrical. Global sea levels might be decreasing as the ice sheets grow but relative sea level around the ice fronts were probably increasing due to glacial isostatic loading. These comments highlight both the importance of this paper and the difficulties that validation faces in terms of the glacial stratigraphic evidence. A partial answer might lie in the use of provenance proxies (e.g. Licht and Hemming, 2017; Verplank et al., 2009; White et al., 2016) in strategically located marine cores from trough-mouth fans or ice sheet proximal deep-sea areas (i.e. Greenland, Labrador, Norwegian Seas). Dating, however, will have to rely on correlations to isotope or paleomagnetic stacks, and the absence of Antarctica in the equation will leave a question mark regardless of the outcome. The authors are to be congratulated on a paper that will give the Quaternary community much food for thought and a very difficult challenge. Some figures are too small.

References cited:

Adam, D. P. 1976. Hudson Bay, Lake Zissaga and the growth of the Laurentide Ice Sheet, Nature, 261, p. 679–680. Andrews, J.T. and Vogt, C. 2020: Variations in felsicversus mafic-sources in the Western Nordic Seas during MIS 1 to MIS 4. Marine Geology, 424, 106164. Clark, P.U. et al., 1993; Initiation and development of the Laurentide and Cordilleran Ice Sheets since the Last Interglacial. Quaternary Sci. Review, 12, 79-114. Dalton, A.S., Finkelstein, S.A., Forman, S.L., Barnett, P.J., Pico, T., and Mitrovica, J., 2019. Was the Laurentide Ice Sheet significantly reduced during Marine Isotope Stage 3? Geology, https://doi.org/10.1130/G45335.1 Jennings, A., Andrews, J., O' Co-
faigh, C., St-Onge, G., Belt, S., Cabedo-Sanz, P., Pearce, C., Hillaire-Marcel, C. and Campbell, C. (2018) Baffin Bay paleoenvironments in the LGM and HS1 : resolving the ice-shelf question', Marine Geology, 402 . pp. 5-16. Miller, G.H. and Andrews, J.T. 2019. Hudson Bay was not deglaciated during MIS-3. Quaternary Science Reviews, Licht, K. J., and Hemming, S. R., 2017, Analysis of Antarctic glacigenic sediment provenance through geochemical and petrologic applications: Quaternary Science Reviews, v. 164, p. 1-24. Refsnider, K.A., Miller, G.H., Fréchette, B., Dylan H. Rood, D.H., 2013. A chronological framework for the Clyde Foreland Formation, Eastern Canadian Arctic, derived from amino acid racemization and cosmogenic radionuclides. Quaternary Geochronology, 16, 21-34. Skinner, R. G., 1973, Quaternary stratigraphy of the Moose River basin, Ontario: Geological Survey of Canada Bulletin 225, 77 p. Simon, Q., Hillaire-Marcel, C., and St-Onge, G. 2012: Late Quaternary chronostratigraphic framework of deep Baffin Bay glaciomarine sediments from high-resolution paleomagnetic data. Geochemistry Geophysics Geosystems DOI: 10.1029/2012GC004272 Simon. Q., Hillaire-Marcel, C., St-Onge, G. and Andrews, J.T. 2014. North-eastern Laurentide, western Greenland and southern Innuitian ice stream dynamics during the last glacial cycle. Journal of Quaternary Science 29(1). DOI: 10.1002/jqs.2648 ć Szabo, B.J., Miller, G.H., Andrews, J.T., Stuiver, M., 1981. Comparison of uranium- series, radiocarbon, and amino acid data from marine molluscs, Baffin Island, Canada. Geology 9, 451-457. Thorleifson, L. H., P. H. Wyatt, W. W. Shilts, and E. Nielsen. 1992. Hudson Bay Lowland Quaternary stratigraphy: evidence for early Wisconsinan glaciation centered in Quebec. in P. U. Clark and P. D. Lea, eds., The Last Interglacial-Interglacial Transition in North America. Geological Society of America Special Paper 270, p. 207-221

Verplanck, E.P., Farmer, G.L., Andrews, J., Dunhill, G., Millo, C., 2009. Provenance of Quaternary glacial and glacimarine sediments along the southeast Greenland margin. Earth and Planetary Science Letters 286, 52-62. White, L.F., Bailey, I., Foster, G.L., Allen, G., Kelley, S.P., Andrews, J.T., Hogan, K., Dowdeswell, J.A., Storey, C.D., 2016. Tracking the provenance of Greenland-sourced, Holocene aged, individual sand-sized

---

## Referee Comment (RC2) · Andrey Ganopolski (Referee) · 31 Mar 2020

The manuscript by Bahadory et al. presents a large ensemble of transient simulations of the last glacial inception performed with a coupled climate-ice sheet model of intermediate complexity. This is an interesting paper which provides a new insight into the mechanisms of glacial inception. However, to be suitable for publication in CP, the manuscript requires a number of clarifications and more critical discussions of potential caveats. Below I describe my major concerns and suggestions.

1. Phase space of last glacial inception

The meaning of the term "phase space of last glacial inception" which authors put in the title and mentioned several times in the text, is unclear to me. Since "phase space" is space, their dimensions (axis) should be properly defined. For example, for mechanical systems, phase space is defined by coordinates and momentum. For the climate system, Fig 3b in Ganopolski et al. (2016) gives an example of another phase space. Here the position of glacial inceptions (bifurcation point) is shown in the insolation–CO2 phase space. The authors should either clearly define what they understand under "phase space" in their manuscript or abandon this term. A similar situation is with the term "bifurcation" which authors used several times (p. 16 and 22) but the meaning of this term remains unclear.

**2. Introduction**

The authors devoted less than one page for discussing previous modelling works related to the last glacial inception. Apart from several own papers, they only cited my publications (Calov et al. (2005); Calov et al. (2009) and Ganopolski et al. (2010)) and the only information Bahadory et al. provide about our works is the spatial resolution of the CLIMBER-2 model: "The model used in that study employed very low resolution (51° longitude by 10° latitude for atmosphere and approximately 100 km for the ice sheet model)" (page 2, line 47). Amazingly, just 20 lines below (page 3, line 66) the authors again describe the spatial resolution of CLIMBER-2: "A further limitation in this latter study is that the CLIMBER EMIC employed uses a 2.5D statistical-dynamical atmospheric model with very limited longitudinal resolution (51.40) and a 3 basin 2D ocean model". Do the authors realize that they describe the resolution of the same model twice? In any case, I believe that the manuscript by Bahadory et al. will benefit a lot if, apart from the resolution of CLIMBER-2, the authors will discuss also other relevant publications. Already in Calov et al. (2005), we cited in the introduction more than 25 modeling papers and since then the number of relevant publications increased significantly.

3. Temperature biases and realistic simulations of ice sheets extent

CPD
The authors stress in the manuscript that they do not use any climate bias corrections and I fully agree that bias correction represents a trade-off between internal consistency and the realism of past climate simulations. All climate models have biases and for simulations of quasi-linear response of the system (like CO2 increase), climate biases are likely not very important. However, for simulation of glacial inception, which is a fundamentally nonlinear process, temperature biases can be much more important because their magnitude can be comparable with the climate response to orbital forcing. In their previous paper (Bahadory and Tarasov, 2018), the authors wrote a lot about temperature biases but provided no information about spatial patterns and magnitude of temperature biases. Table 3 only indicates that average temperature over the box covering most of Canada is close to reality. The real problem is, however, not average but strong (5 to 10oC depending on the season) zonal temperature gradient over northern North America related to the atmospheric circulation and explained by quasi-stationary planetary waves. Due to the coarse spatial resolution of CLIMBER-2. this effect is not resolved and this leads to a strong, dipole-like temperature bias (Ganopolski et al., 2010; Fig 2a). This is why it is noteworthy that the North American ice sheets simulated without temperature bias correction in Calov et al. (2005a) (Fig. 6) and in Bahadory et al. (Fig. 4 and 5) are very similar with the thickest ice located over Alaska. Introducing of temperature bias correction in Ganopolski et al. (2010) led to a very different ice sheet evolution which we believe is more realistic. The similarity between Bahadory et al. and our old results (Calov et al., 2005) can be caused by the fact that the LOVECLIM model, in spite of a higher spatial resolution, has a rather simplistic atmospheric model which results in similar to CLIMBER-2 temperature biases. At least, this is what one can conclude from Fig. 1b in Heinemann et al. (2014), another paper based on the LOVECLIM model. By the way, in this paper temperature bias correction has been used. Bias correction has been used also in a number of GCMs studies such as Vizcaino et al. (2008); Herrington and Poulsen (2012). This is why, it would be useful to show present-day (preindustrial) summer temperature biases simulated by LOVECLIM model used by Bahadory et al. This can be, for example,
the average value over the 50 ensemble members or a single representative one. Of course, it is up to the authors to decide which technique to employ but they should inform their readers about potential serious drawbacks.

**4. Present-day constraints on model parameters**

On page 10 the authors wrote that "despite having different start times (due to different calendar start years between 122 ka and 119 ka ...), all simulations start growing ice in the first 100 years of simulation". It is not clear from the paper which runs started at which time, as well as why start time was chosen differently for different runs. However, the fact that according to Fig. 1 the model simulates between 10 to 20 meters sea level drop already at 119 ka is worrisome. Indeed, since climate before 120 ka was similar to preindustrial one or even warmer, such rapid ice sheet growth at the beginning of model runs indicates that at least some model realizations would simulate glacial inception already during the late Holocene which, of course, is in odd with observational data. In Section 5, Bahadory and Tarasov (2018) wrote that they used "a trial criteria based on ice volume changes (between 1700 and 1980 CE)" to reject model versions which grow "too much" ice during this interval. But this interval is much too short for such a test. For example, Fig.3 in Bahadory et al. clearly shows how much ice is formed after year 200 since the beginning of the runs. Since summer insolation and GGHs concentrations remained practically constant at least since 1000 BCE till ca. 1900 CE, testing of whether or not selected model versions simulate glacial inception in the late Holocene would require at least 10 times longer runs than have been performed by the authors. To be able to judge their realism, it is crucially important to know how much "present-day" ice is simulated by different model versions.

5. Spatial patterns of simulated North American ice sheet

When discussing spatial patterns of simulated Laurentide ice sheet, the authors wrote "to our knowledge, there is no community-based geologically-inferred MIS 5 ice margin reconstruction for NA. Aside from the issue of Alaska (and certain adjacent parts of the
Yukon), our results are, within (large) age uncertainties, consistent with the till stratigraphy presented in Clark et al. (1993)" (page 22). However, Clark et al (1993) explicitly stated that "the Laurentide Ice Sheet first developed during Stage 5 over Keewatin, Quebec and Baffin Island" (page 79) which is inconsistent with the results presented by the authors. It is also noteworthy that the recent reconstruction of NH ice sheet for the MIS 5d presented in Batchelor et al. (2018), also places MIS 5d Laurentide ice sheet over northern-eastern Canada and implies very little glaciation over Alaska and in Western Canada. Since I am not an expert in the history of glaciation of North America during the last glacial inception, I wonder what the authors think about these apparent inconsistencies? And if they really believe that there are no reliable reconstructions for the ice sheets during MIS 5d (I do not understand the meaning of "community-based"), then what is the motivation for performing a large ensemble of transient last glacial inception simulations?

6. Using simulations of glacial inceptions to constrain transient climate response

The authors devoted only one paragraph in the text to the description of how they used their model ensemble to constrain transient climate response (TCR). However, they highlighted this result in the abstract where they suggested that their results can be used "to constrain future climate change". Since future climate change is a very hot issue, this small part of the manuscript deserves serious attention, especially, because the authors put a very tight constraint on TCR (0.7-1.4 C). If their estimate of TCR is correct, then only five of ca. 30 different GCMs participating in CMIP5 have the right TCR while all other overestimate it. Obviously, such a claim has very serious implications for future climate change projections. Below I argue why simulations of glacial inception cannot constrain future climate change.

Although a number of attempts have been made to use paleoclimate data and results of paleoclimate modelling to constrain equilibrium climate sensitivity (ECS), these studies cannot directly constrain TCR. Indeed, although there is some correlation between TCR and ECS, this correlation is not very tight and TCR of different models with similar
ECS can differ by factor two. The reason is that TCR strongly depends on the rate of ocean heat uptake which differs significantly between climate models. Obviously, simulations of glacial inception provide no constraints on ocean heat uptake. This is why below I only discuss whether simulations of glacial inception can constrain ECS.

i) Climate sensitivity to CO2 doubling (ECS) and the response of climate to seasonal and latitudinal redistribution of insolation are caused by completely different forcings an, therefore, numerous processes and feedbacks play a completely different role. I am not aware of any study about the relationship between regional and seasonal climate response to insolation change and global climate response to CO2 change (ECS), but I doubt whether there is a strong correlation between these very different climate changes.

ii) As far as the simulated rate of ice sheet growth is concerned, the situation is even more complex because ice sheet growth is controlled not only by simulated climate change but by many other factors. The first one is model biases in modern climatology. If these biases are comparable with climate response to orbital forcing, then there is a big question of whether ice sheets growth can constrain future climate change. Second, ice sheet response to orbital forcing strongly depends on surface mass balance parameterization. The authors used the PDD scheme which does not even explicitly account for the direct impact of orbital forcing on the surface mass balance of ice sheets and a number of studies (e.g. van de Berg et al., 2011; Bauer and Ganopolski, 2017) questioned the applicability of this simplistic scheme to the modelling of ice sheet response to orbital forcing.

In short, I do not believe that simulations of glacial inception can really constrain ECS, let alone TCR.

References (not cited in the manuscript by Bahadory et al.)

Batchelor, C.L., Margold, M., Krapp, M., Murton, D.K., Dalton, A.S., Gibbard, P.L., Stokes, C.R., Murton, J.B. and Manica, A., 2019. The configuration of Northern Hemi-

CPD
sphere ice sheets through the Quaternary. Nature communications, 10, 1-10, 2019. Bauer, E. and Ganopolski, A.: Comparison of surface mass balance of ice sheets simulated by positive-degree-day method and energy balance approach, Clim. Past, 13, 819–832, 2017. Ganopolski, A., Winkelmann, R., Schellnhuber, H.J. Critical insolation-CO2 relation for diagnosing past and future glacial inception. Nature, 529, 200-203, 2016. Heinemann, M., A. Timmermann, O. Elison Timm, F. Saito, and A. Abe-Ouchi. Deglacial ice sheet meltdown: orbital pacemaking and CO2 effects, Clim. Past, 10, 1567–1579, 2014. van de Berg, W. J., van den Broeke, M., Ettema, J., van Meijgaard, E., and Kaspar, F.: Significant contribition of insolation to Eemian melting of the Greenland ice sheet, Nat. Geosci., 4, 679–683, 2011.

---

## Author Comment (AC1) · 16 Jun 2020

We thank both reviewers for thoughtful/constructive responses.

**Response to review/comments by John Andrews**

*But for many readers the important point that*
*this paper make is in section 4.4 "Brief comparison to past geological inferences"*
*is indeed brief a mere 8 lines but this statement outlines the other important, indeed*
*critical, verification of the modelled ice sheets and their expansion and retraction, that*
*is the aerial extent of the ice sheets, a necessary but not sufficient parameter in the*
*calculations of ice volume an global sea level. It is a call for action to the glacial geological*
*community, however, the problems have not changed significantly since the Clark*
*et al 1993 paper—that is the ability to provide a date on buried stratigraphic units,*
*primarily tills, that are older than the 50,000 radiocarbon dating limit—this problem*
*remains.*

Yup, very brief. The intent, as noted, is to prompt the glacial geology community with
some model-based chronologies. Some of the reviewer's comments will in part be incorporated/addressed
into the revised text but we will also explicitly refer the reader to this review for the valuable
discussion and to ensure appropriate academic credit.

*so it is difficult to*
*see why the growth and decay should be more symmetrical.*

Whether the saw-tooth paradigm is appropriate for shorter stadial/interstadial
transitions is unclear to us. Yes a large LGM NA ice complex with extensive warm
based regions should have fast and strong deglacial intervals, but whether this
is necessary for MIS5d is unclear. The model used in this study, does lack shallow-shelf
approximation ice physics (now addressed in ongoing work), so grounding line retreat
is poorly represented. But the terrestrial components should have more confidence. The
revised submission will give more guidance on model uncertainties and how they should
be taken into account by readers.

*Some figures are too small.*

Will address in revision.

**Response to review by Andre Ganopolski**

1. Phase space of last glacial inception

*The meaning of the term "phase space of last glacial inception" which*
*authors put in the title and mentioned several times in the text, is*
*unclear to me. Since "phase space" is space, their dimensions (axis)*
*should be properly defined. For example, for mechanical systems, phase*
*space is defined by coordinates and momentum. For the climate system,*
*Fig 3b in Ganopolski et al. (2016) gives an example of another phase*
*space. Here the position of glacial inceptions (bifurcation point) is*
*shown in the insolation–CO2 phase space. The authors should either*
*clearly define what they understand under "phase space" in their*
*manuscript or abandon this term. A similar situation is with the term*
*"bifurcation" which authors used several times (p. 16 and 22) but the*
*meaning of this term remains unclear.*

Fair enough for some sloppy useage. We clearly aren't using the
standard physics definition for phase space.  We are debating between
replacing with "trajectory space": or defining phase space as the "4D

space of possible ice/climate histories over glacial inception as represented by LCice". Bifurcation has been used in the popular sense of division into disjoint branches with respect to some characteristic (and not in the technical sense of dynamics systems theory). We'll add a footnote clarifying this.

2. Introduction

*The authors devoted less than one page for discussing previous modelling works related to the last glacial inception. Apart from several own papers, they only cited my publications (Calov et al. (2005); Calov et al. (2009 ) and Ganopolski et al. ( 2010)) and the only information Bahadory et al. provide about our works is the spatial resolution of the CLIMBER-2 model: "The model used in that study employed very low resolution (51longitude by 10 latitude for atmosphere and approximately 100 km for the ice sheet model)" (page 2, line 47).*

For the record, the claim is incorrect. Our submission also states "the one coupled ice/climate modelling study that adequately captured both the growth and retreat phases of LGI required the use of an imposed (albeit plausible) aeolian dust deposition forcing and temperature bias correction (Ganopolski et al., 2010)"

*"On other hand, it should be noted that the relative quality of modelled LGM ice extent in Ganopolski et al. (2010) attests the potential value of using fast EMICS like CLIMBER"*

*Amazingly, just 20 lines below (page 3, line 66) the authors again describe the spatial resolution of CLIMBER-2: "A further limitation in this latter study is that the CLIMBER EMIC employed uses a 2.5D statistical-dynamical atmospheric model with very limited longitudinal resolution (51.4o) and a 3 basin 2D ocean model". Do the authors realize that they describe the resolution of the same model twice?*

Oops, sorry about that. Now fixed. (Submission was rushed to make the IPCC deadline, based on North American time...)

*In any case, I believe that the manuscript by Bahadory et al. will benefit a lot if, apart from the resolution of CLIMBER-2, the authors will discuss also other relevant publications. Already in Calov et al. (2005), we cited in the introduction more than 25 modeling papers and since then the number of relevant publications increased significantly.*

We agree that the submission would benefit from more review of past work, and will do so (especially for more recent papers).  However, we also note that some of the "25 modeling papers" cited in Calov et al. (2005) used what we judge to be poor model/experimental configuration/designs and obtained poor results in large discord with paleo proxy constraints. Some also just used flow-line models and/or otherwise lacked 2D geographic resolution. For these cases, we see no point in referencing.

3. Temperature biases and realistic simulations of ice sheets extent

*The authors stress in the manuscript that they do not use any climate bias corrections and I fully agree that bias correction represents a trade-off between internal consistency and the realism of past climate simulations. All climate models have biases and for simulations of quasi-linear response of the system (like CO2 increase), climate biases are likely not very important. However, for simulation of glacial inception, which is a fundamentally nonlinear process, temperature biases can be much more important because their magnitude can be comparable with the climate response to orbital forcing. In their previous paper (Bahadory and Tarasov, 2018), the authors wrote a lot about temperature biases but provided no information about spatial patterns and magnitude of temperature biases. Table 3 only indicates that average temperature over the box covering most of Canada is close to reality. The real problem is, however, not average but strong (5 to 10oC depending on the season) zonal temperature gradient over northern North America related to the atmospheric circulation and explained by quasi-stationary planetary waves. Due to the coarse spatial resolution of CLIMBER- 2, this effect is not resolved and this leads to a strong, dipole-like temperature bias (Ganopolski et al., 2010; Fig 2a). This is why it is noteworthy that the North American ice sheets simulated without temperature bias correction in Calov et al. (2005a) (Fig. 6) and in Bahadory et al. (Fig. 4 and 5) are very similar with the thickest ice located over Alaska. Introducing of temperature bias correction in Ganopolski et al. (2010) led to a very different ice sheet evolution which we believe is more realistic. The similarity between Bahadory et al. and our old results (Calov et al. , 2005) can be caused by the fact that the LOVECLIM model, in spite of a higher spatial resolution, has a rather simplistic atmospheric model which results in similar to CLIMBER-2 temperature biases. At least, this is what one can conclude from Fig. 1b in Heinemann et al. (2014), another paper based on the LOVECLIM model. By the way, in this paper temperature bias correction has been used. Bias correction has been used also in a number of GCMs studies such as Vizcaino et al. (2008); Herrington and Poulsen (2012). This is why, it would be useful to show present-day (preindustrial) summer temperature biases simulated by LOVECLIM model used by Bahadory et al. This can be, for example, the average value over the 50 ensemble members or a single representative one. Of course, it is up to the authors to decide which technique to employ but they should inform their readers about potential serious drawbacks.*

Good point. And as seen below, the NA present-day biases are large for Loveclim over the range of models that passed our acceptance threshold for glacial inception.  However, even with the large present-day warm bias, Baffin is one of the first places to glaciate in our model.

Eurasian biases are relatively much less (not shown, but will be in the revised submission).  We will be adding a much more complete discussion on model limitations and how this should affect interpretation of results. This will include plots of subensemble means and variances of present-day bias.

[Figure]

Figure: Passed subensemble JJA mean surface air temperature bias.

[Figure]

Figure: passed subensemble standard deviation JJA surface air temperature

**4. Present-day constraints on model parameters**

*On page 10 the authors wrote that "despite having different start times (due to different calendar start years between 122 ka and 119 ka ...), all simulations start growing ice in the first 100 years of simulation". It is not clear from the paper which runs started at which time, as well as why start time was chosen differently for different runs. However, the fact that according to Fig. 1 the model simulates between 10 to 20 meters sea level drop already at 119 ka is worrisome. Indeed, since climate before 120 ka was similar to preindustrial one or even warmer, such rapid ice sheet growth at the beginning of model runs indicates that at least some model realizations would simulate glacial inception already during the late Holocene which, of course, is in odd with observational data. In Section 5, Bahadory and Tarasov (2018) wrote that they used "a trial criteria based on ice volume changes (between 1700 and 1980 CE)" to reject model versions which grow "too much" ice during this*

*interval. But this interval is much too short for such a test. For example, Fig.3 in Bahadory et al. clearly shows how much ice is formed after year 200 since the beginning of the runs. Since summer insolation and GGHs concentrations remained practically constant at least since 1000 BCE till ca. 1900 CE, testing of whether or not selected model versions simulate glacial inception in the late Holocene would require at least 10 times longer runs than have been performed by the authors. To be able to judge their realism, it is crucially important to know how much "present-day" ice is simulated by different model versions.*

We respectivefully disagree on the present-day test interval being to short. The interval was appropriate for the given context of extracting an ensemble of 500 model runs closest to equilibrium mass-balance out of 2000 model runs.

As to the question of whether the models are positive mass-balance biased, that is already clearly the case for most models from figure 10 of Bahadory and Tarasov (2018). The more fundamental question of to what extent this distorts the results of the present work is in good part answered by the result of having models that subsequently retreat post-stadial at a rate that is consistent with sealevel proxies (within relevant proxy uncertainties).

The Northern mid to high latitude Eemian summer insolation maximum occured around 126 ka and Lisieki/Raymo 2004 have their Eemian sealevel highstand (likely dominated by Antarctica) at 123 ka. So it may well be that North America started growing ice earlier than the 120-122 ka our model runs started at. As such, the late start time likely offsets some (or all?) of the impact of the present-day positive surface mass-balance bias.

The reviewer also raises the point that start times for the passed subensemble don't have their actual start times listed. We will add a list of the subensemble parameter vectors to the revised supplement.

5. Spatial patterns of simulated North American ice sheet

*When discussing spatial patterns of simulated Laurentide ice sheet, the authors wrote "to our knowledge, there is no community-based geologically-inferred MIS 5 ice margin reconstruction for NA. Aside from the issue of Alaska (and certain adjacent parts of the Yukon), our results are, within (large) age uncertainties, consistent with the till stratigraphy presented in Clark et al. (1993)" (page 22). However, Clark et al (1993) explicitly stated that "the Laurentide Ice Sheet first developed during Stage 5 over Keewatin, Quebec and Baffin Island" (page 79) which is inconsistent with the results presented by the authors. It is also noteworthy that the recent reconstruction of NH ice sheet for the MIS 5d presented in Batchelor et al. (2018), also places MIS 5d Laurentide ice sheet over northern-eastern Canada and implies very little glaciation over Alaska and in Western Canada. Since I am not an expert in the history of glaciation of North America during the last glacial inception, I wonder what the authors think about these apparent inconsistencies?*

*And if they really believe that there are no reliable reconstructions for the ice sheets during MIS 5d (I do not understand the meaning of "community-based"), then what is the motivation for performing a large ensemble of transient last glacial inception simulations?*

Clark et al. (1993) do not discuss Ellesmere and they differentiate Laurentide from the Cordilleran ice sheet. As for the Western Canadian Arctic, their discussion is summarized in fig 16, which indicates no constraints during MIS5d:c. Furthermore, their summary figure 16 shows no MIS5d glaciation over Quebec, contradicting the statement in the abstract. So the above quote from Clark et al, in combination with their figure 16 is consistent with the results in our fig 3 and fig 5. We will add some of these details to the discussion in the revision.

We would also argue that Batchelor et al. (2018) best estimate MIS5e in their figure 1 is inconsistent with sealevel constraints. Their use of a single scaling estimate for ice volume is not appropriate when you have multiple domes nor an ice sheet that is unlikely to be in equilibrium. 3 circular ice caps of the same total area as a single circular ice cap will have less total ice volume. Discrepancies can get even larger when you use have non-circular geometries. We therefore find this reconstruction problematic, but will look more carefully at their cited constraints, to see what aspects are more robust and address this in the revisions.

6. Using simulations of glacial inceptions to constrain transient climate response

*The authors devoted only one paragraph in the text to the description of how they used their model ensemble to constrain transient climate response (TCR). However, they highlighted this result in the abstract where they suggested that their results can be used "to constrain future climate change". Since future climate change is a very hot issue, this small part of the manuscript deserves serious attention, especially, because the authors put a very tight constraint on TCR (0.7-1.4 C). If their estimate of TCR is correct, then only five of ca. 30 different GCMs participating in CMIP5 have the right TCR while all other overestimate it. Obviously, such a claim has very serious implications for future climate change projections.*

The reviewer should have quoted the full sentence from the abstract: "This therefore underlines the potential value of fully coupled ice/climate modelling of last glacial inception to constrain future climate change". IE, we do not claim that our modelling results should be used to constrain climate sensitivity only that there is potential value to do so from fully coupled ice/climate modelling.

*Below I argue why simulations of glacial inception cannot constrain future climate change. Although a number of attempts have been made to use paleoclimate data and results of paleoclimate modelling to constrain equilibrium climate sensitivity (ECS), these studies cannot directly constrain TCR. Indeed, although there is some correlation between TCR and ECS, this correlation is not very tight and TCR of*

*different models with similar ECS can differ by factor two. The reason
is that TCR strongly depends on the rate of ocean heat uptake which
differs significantly between climate models. Obviously, simulations
of glacial inception provide no constraints on ocean heat uptake. This
is why below I only discuss whether simulations of glacial inception
can constrain ECS.*

*i) Climate sensitivity to CO2 doubling (ECS) and the response of
climate to seasonal and latitudinal redistribution of insolation are
caused by completely different forcings an, therefore, numerous
processes and feedbacks play a completely different role. I am not
aware of any study about the relationship between regional and
seasonal climate response to insolation change and global climate
response to CO2 change (ECS), but I doubt whether there is a strong
correlation between these very different climate changes.*

*ii) As far as the simulated rate of ice sheet growth is concerned, the
situation is even more complex because ice sheet growth is controlled
not only by simulated climate change but by many other factors. The
first one is model biases in modern climatology.  If these biases are
comparable with climate response to orbital forcing, then there is a
big question of whether ice sheets growth can constrain future climate
change. Second, ice sheet response to orbital forcing strongly depends
on surface mass balance parameterization. The authors used the PDD
scheme which does not even explicitly account for the direct impact of
orbital forcing on the surface mass balance of ice sheets and a number
of studies (e.g. van de Berg et al., 2011; Bauer and Ganopolski, 2017)
questioned the applicability of this simplistic scheme to the
modelling of ice sheet response to orbital forcing.*

There is no "the PDD scheme", and the scheme we use (temperature
dependent melt coefficients derived from energy balance modelling) is
different than what most have used to date and arguably indirectly
does take into account SW dependencies better than the common PDD
scheme with fixed degree-day melt coefficients. It still though does
not have explicit dependence on solar insolation and we will make this
caveat explicit in the revised text. It should also be noted, that a
surface mass-balance scheme with explicit surface insolation
dependence has been implemented in the GSM in 2019 (and is now the default, but
came too late for the ensembles in this project).

*In short, I do not believe that simulations of glacial inception can
really constrain ECS, let alone TCR*

This is a fair critique (and what should be be obvious in hindsight
learning on TCR versus ECS) and we have now computed the ensemble ECS
for comparison (and will replace TCR with ECS examination in the
revisions). As per the figure below, the requirement of capture of
last glacial inception and subsequent retreat still provides some
constraint on ECS, rejecting runs with ECS < 1.3 C. But now this
criteria does not provide an upper bound constraint, in contrast to
that for TCR. However, this figure also shows the limited range of ECS
probed over the current ensemble (which us being addressed in ongoing work).
ECS will depend on the radiative forcing of 2*CO2 (which varies
somewhat across models) as well on internal feedbacks. The reviewer

fairly points out that response to orbital changes in insolation will
be subject to different feedbacks than that for future 2*CO2. However,
some of these feedbacks will be similar (eg snow and sea-ice albedo).
Furthermore, last glacial inception also included changes in pCO2.

There is also a submission (Choudhury, Simulating Marine Isotope Stage
7 with a coupled climate-ice sheet model) that recently completed TCD
discussion of relevance. It used LoveClim radiative forcing dependence
on CO2 as one of its two ensemble parameters. Again, capture of the
stadial/interstadial response strongly narrowed down parameter
ranges. For this case, I would therefore expect a strong correlation
between model ECS and stadial/interstadial capture.

ECS test of
full
ensemble
(0ka to
3ka).
Simulations
using
parameter
vectors that
passed
glacial
inception
constraints
are in black.

---

## Author Response (AR2)

**Response to the reviewers**

We again thank the reviewer, Andrey Ganopolski, for taking time for this final? review.
* * *
**Reviewer 1, Andrey Ganopolski**

**Reviewer Point P 1.1** —

I appreciate the efforts which the authors made to respond to my criticism and suggestions and in general, I am satisfied with their response. However, I still have two remarks on the revised manuscript.

1) In response to my criticism that in the introduction the authors do not cite previous works on modelling of glacial inception (nothing personal – all my relevant papers have been cited in the manuscript by Bahadory et al.), the authors responded " that most of the "25 modeling papers" cited in Calov et al. (2005) used what we judge to be "poor model/experimental configuration/designs and obtained poor results in large discord with paleo proxy constraints ..." Since many more papers on this subject have been published after Calov et al. (2005), the authors obviously consider all of them also to be "poor". I do not believe that such attitude (unfortunately, not unusual) when previous publications considered to be obsolete and not worth mentioning just because they were based on the "wrong" models. However, only thanks to these earlier studies, we now can do some things better than it was possible 15 or 20 years ago and this is why the earlier efforts deserve at least to be acknowledged. After all, do the authors believe that their own study is problems-free and their results are in perfect agreement with paleo proxy constraints?

**Reply**: By Andrey's logic one might argue that every published say GCM study should cite every every previous GCM study for the given context, which would make the papers unreadable and the majority of text plain citation. Blind citation does not help the reader in our opinion. Yes, we acknowledge citing early ground-breaking work, but only to a point. Furthermore, a number of the early papers that Andrey cites in his 2010 paper are not directly relevant to this study, ie exploring the possible to likely geographic evolution of the last glacial inception ice sheets. We have, though, done one more near exhaustive litterature source on the topic, including all papers that cite Calov et al. (2005) (according to web of science). And have added the following 3 citations: Bonelli et al., 2009, Herrington and Poulsen (2011), Gregory et al. (2012. We've also added a whole paragraph about coupled GCM/ice sheet modelling of LGI that references the latter two.

*many more papers on this subject have been published after Calov et al. (2005), the authors obviously consider all of them also to be*

There are not many that have modelled northern hemispheric last glacial inception with fully coupled models that offer some probability of encapsulating the geographic evolution of the ice sheets (as opposed to earlier studies that just focused on explaining ice volume changes inferred from sea level proxies). And we see no point in citing coupled modelling studies that examined different glacial intervals.

**Reviewer Point P 1.2** — 2) In my first review, I suggested that the modelled east-west asymmetry in the North American ice sheet distribution is not consistent with paleoclimate reconstructions and is caused by temperature biases of the LOVECLIM model. Following my recommendation, the authors now show (Fig. 14) modelled summer temperature biases and discuss their potential

implications for modelling of glacial inception in section 3.1.2 ("Labrador and eastern NA remain ice-free, likely due to warm model biases in this region") and 4.1 ("... temperature biases ... may have inhibited glaciation over Hudson Bay and northern Quebec"). However, in section 4.5, the authors continue to insist that ("except for Alaska") their modelling results are consistent with paleoclimate reconstructions.

**Reply**: The last statement is incorrect. We state "Except for Alaska (and certain adjacent parts of the Yukon), our results are, within (large) age uncertainties, consistent with the till stratigraphies presented in Clark et al. (1993) and their summarizing figure 19", and that statement is correct given the large uncertainties in ages and incomplete statigraphic sequences.

**Reviewer Point P 1.3** — To this end, the authors dismiss MIS5d reconstruction by Batchelor et al. (2018) as unreliable and claim that what Clark et al. (1993) wrote in their paper about the initialization of glaciation in Keewatin and Quebec contradicts to their own figure (Fig. 19) which shows the opposite. Although the authors are much closer to Quebec than me, I would insist that their reading of Fig. 19 in Clark et al. (1993) is mistaken. Clark et al. (1993) wrote "Following the last interglaciation, the Laurentide Ice Sheet first developed during Oxygen-Isotope Stage 5 over Keewatin, Quebec and Baffin Island" on the same page where they discuss Fig. 19. This is why I doubt that the text and Fig. 19 contradict each other. In fact, Fig. 19 only suggests that ice sheet was absent in the most southern part of Quebec during MIS5 which does not contract, for example, Batchelor's MIS5d reconstruction.

**Reply**: As was indicated in the revised submission, Batchelor et al, in their supplement list available empirical data points for each of their time-slices. There are none for MIS5d Laurentide. The sole constraint (apart from one small region) is the empirical reconstruction from Kleman et al, 2010, based on glacial flow indicators (flow sets). Chronology is again a major challenge, and the latter are unable to rule out that the indicators used were for pre-Eemian flow: " If older than the Wisconsinan, such restricted ice volumes are only compatible with a stage preceding the Illinoian maximum, an alternative we consider less probable due to the preservation of the Atlantic morphology." For which I would counter, how was this morphology preserved through MIS3:1?

We also now bring up the contradiction between Batchelor's MIS5d LIS and Clark et al. (1993), with one having complete Hudson Bay glaciation and the other having it ice free:

"Their complete lack of MIS5d glaciation of Hudson Bay contradicts the favoured inference of Clark et al. (1993), 470 pointing to the challenge of inferences from sparse geological data with very poor age control. It should be noted however that Batchelor et al. (2019) barely reach the inferred MIS5d sea level minimum of ?, using an ice volume to area scaling relationship derived for a circular mono-dome ice sheet with plastic rheology (Cuffey and Paterson, 2010). We find that this relationship over-estimates ice volume during last glacial inception by at least 50% when compared to the base GLAC1-D nn9927 ice sheet chronology from the calibrated glaciological modelling of Tarasov et al. (2012)."

Is this not a major feature that raises questions about which of these are accurate? :

I (Lev Tarasov) have reviewed Clark et al 2003 one more time. The core issue is that no data is provided for Quebec and Labrador aside from the chronologies in fig 11 and 12 (St. Lawrence Lowlands and Appalachian uplands), which indicate an absence of glacial tills over the 75 ka (option A) or 95 ka (option B) to 130 ka interval.

The Clark et al 1993 reasoning for the statements that Andrey is referring to relate to (as far as I can tell) the following quote "The Rocksand and Amery tills underlie the Fawn River and Nelson River

sediments, respectively (Fig. 17). The Rocksand Till contains reworked marine shells with alle/Ile values similar to those obtained from Bell Sea sediments (Wyatt, 1989). If the TL chronology is correct, these tills record west-northwestward flow of ice from Quebec during the middle of Oxygen-Isotope Stage 5" (Clark et al 1993). However the TL chronologies are quite uncertainty.

Anyway, to ensure my interpretation is accurate, I contacted two of the co-authors of Batchelor et al, 2016. Both are glacial geologists who have worked extensively on last glacial cycle records for North America. They indicated:

(Chris Stokes, Durham U., pers. comm.) "I don't really see Kleman et al.'s work as much of a constraint. It's a sketchy bit of evidence that is undated. The problem is that it is pretty much all we have." It should be noted, that this cited Kleman et al, 2010 work was virtually the sole input for the Batchelor et al, minimum and best guess Laurentide ice sheet configurations for MIS5d.

(April Dalton, Durham U., pers. comm.) "As for the Hudson Bay Lowlands, the stratigraphic record there is complex and unfortunately its not currently possible to determine whether this area was ice-free or glaciated during MIS 5 stadials." As noted above, it is these records that form the basis for the Clark et al (1993) interpretation of early formation of ice over Quebec and Labrador.

Given all this, we've rewritten the NA comparison against past geological inferences, to better emphasize the uncertainties. We've also removed that statement comparing to Clark et al 1993 that Andrey finds so problematic and now refer to a more recent (though shorter review, but at least which presents/cites more useful data about Quebec LGI):

"Stokes et al. (2012a) provides the most recent review of geological inferences and modelling results for NA LGI. The main discrepancies in our results are the already noted issue of excessive Alaskan glaciation and likely inadequate ice extent over Quebec and Labrador."

**Reviewer Point P 1.4** —

In any case, whatever the reliability of paleoclimate reconstructions of MIS5 North America ice sheet is, the Fig. 14 in the revised Bahadory et al. manuscript shows that their modelling results in the north-eastern part of North America are not trustworthy. Indeed, the figure shows summer temperature biases over Keewatin and Quebec of more than +10C (!). Since, according to model simulations, climate response to changes in orbital forcing at 116ka was only half of that, this area in the model remains much warmer during glacial inception than it is in reality even at present. This, of course, completely preclude ice sheet formation in eastern Canada. As I made it clear in my previous review, I have no intention to give advice to the authors which modelling approach they should use. However, I would now add to the list of publications which applied temperature biases corrections, the recent paper by Choudhury et al. (2020) and I believe this paper reinforces my concern about the impact of strong summer temperature biases in LOVECLIM on simulation of glacial inception.

**Reply**: We already state "Model temperature biases under present-day conditions are larger over NA than over EA and may have inhibited glaciation over Hudson Bay and northern Quebec". And it should be noted that Loveclim has a stronger warm bias over Hudson Bay then the adjacent sector (ie same latitude) of Quebec/Labrador, but that doesn't inhibit full glaciation of Hudson Bay (though not James Bay) in LCice by 112 ka. However, to further assuage Andrey's concerns, and ensure that the readers are clear on simulation limitations, we have somewhat rewritten that subsection

"Figure 14 shows the present-day mean June/July/August temperature bias of the 55-member ensemble for both NA and EA. Model temperature biases under present-day conditions are larger over NA than over EA and the 9:5oC to 14oC mean regional summer temperate biases would have hindered

at least initial glaciation and likely LGI maximum extent (to a more uncertain degree) over Hudson Bay, Quebec, and Labrador. However, though the ensemble mean summer temperature bias is stronger 410 over Hudson Bay than the adjacent sector of Quebec/Labrador, this doesn't preclude complete glaciation over Hudson Bay (but not James Bay) by 112 ka (figure 6) in ensemble members. Increased LGI stadial ice over these latter regions would also improve fits to global mean sea-level proxies (cf. Figure 1). Thicker stadial ice could also enable a stronger and faster post-stadial retreat."

and now also make explicit reference to the issue in the conclusions:

"Discrepancies are likely due to the absence of a modelled (and probably out of phase) Antarctic ice sheet contribution in LCice 1.0, model limitations (as evidenced by the present-day warm summer temperature bias over Hudson Bay and Quebec), and dating uncertainties in the proxy-based reconstructions"